# It Takes Two to Tango: Directly Optimizing for *Constrained* Synthesizability in Generative Molecular Design

## Abstract

Constrained synthesizability is an unaddressed challenge in generative molecular design. In particular, designing molecules satisfying multi-parameter optimization objectives, while simultaneously being synthesizable *and* enforcing the presence of specific building blocks in the synthesis. This is practically important for molecule re-purposing, sustainability, and efficiency. In this work, we propose a novel reward function called **TANimoto Group Overlap (TANGO)**, which uses chemistry principles to transform a sparse reward function into a *dense* reward function – crucial for reinforcement learning (RL). TANGO can augment molecular generative models to *directly* optimize for constrained synthesizability while simultaneously optimizing for other properties relevant to drug discovery. Our framework is general and addresses starting-material, intermediate, and divergent synthesis constraints. Contrary to many existing works in the field, we show that *incentivizing* a general-purpose model with RL is a productive approach to navigating challenging synthesizability optimization scenarios. We demonstrate this by showing that the trained models explicitly learn a desirable distribution. Our framework is the first *generative* approach to successfully address constrained synthesizability. The code is provided at https://figshare.com/s/0aa1bb23734ee16d4331.

## 1. Introduction

Synthesizable generative molecular design is becoming increasingly prevalent (Gao & Coley, 2020; Stanley & Segler, 2023), paralleling the rise in the number of experimentally validated generative design case studies (Du et al., 2024).

Controlling *how* generated molecules can be synthesized offers great potential for the push towards closed-loop discovery (Coley et al., 2020a;b) as molecules that can be made from specific reagents or reactions are naturally more amenable to robotic synthesis automation, which can be specialized for certain chemistries (Tom et al., 2024; Strieth-Kalthoff et al., 2024; Sin et al., 2024). Moving beyond methods that optimize for synthesizability heuristics (Stanley & Segler, 2023; Neeser et al., 2023), approaches that *explicitly* assess synthesizability can be broadly categorized into forward- or retro-synthesis which builds molecules from simple building blocks, or recursively decomposes a target molecule into constituent building blocks, respectively. An example of forward-synthesis in the context of molecular design is synthesizability-constrained molecular generation. These methods anchor molecular generation in viable chemical transformation rules, thus *promoting* synthesizability (Gao et al., 2022; 2024). On the other hand, retrosynthesis planning (Liu et al., 2017; Segler & Waller, 2017; Coley et al., 2017; Segler et al., 2018) proposes viable synthetic routes to a target molecule, and these models are often used as stand-alone tools to assess synthesizability. Such models have become increasingly adopted and are now routinely used to filter generated molecules (Shields et al., 2024). Recent work has shown that generative models can *directly* generate molecules deemed synthesizable by retrosynthesis models by treating them as another oracle (computational prediction) to optimize for (Guo & Schwaller, 2024c). Subsequently, *constrained* synthesis planning has become a research focus, whereby proposed synthetic routes incorporate *enforced building blocks*. This is especially relevant for sustainability and efficiency and examples include semi-synthesis (Vollmann et al., 2022) (start from reagents isolated from natural sources) and divergent-synthesis (Li et al., 2018) (pass through common intermediates). More examples include starting-material constrained synthesis (Granda et al., 2018; Wołos et al., 2020), which can also re-purpose waste to valuable molecules (Wołos et al., 2022; Żądło-Dobrowolska et al., 2024). More recently, constrained retrosynthesis algorithms have been proposed (Johnson et al., 1992; Yu et al., 2022; 2024). However, to date, there are no molecular generative models that can enforce specific building blocks in the proposed routes.

[1]Anonymous Institution, Anonymous City, Anonymous Region, Anonymous Country. Correspondence to: Anonymous Author <anon.email@domain.com>.

Preliminary work. Under review by the International Conference on Machine Learning (ICML). Do not distribute.

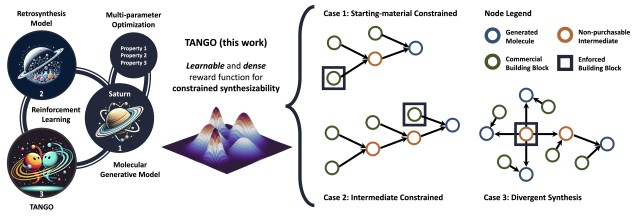

*Figure 1.* TANGO guides the generation of molecules directly optimized for constrained synthesizability with *enforced building blocks* while simultaneously optimizing other properties. Our method generalizes across starting-material, intermediate, and divergent synthesis constraints.

In this work, we show that a *general-purpose* molecular generative model, without any constraints, can be *incentivized* to generate synthesizable molecules that satisfy multi-parameter optimization (MPO) objectives while jointly *enforcing* a set of building blocks. Our contribution is as follows: **(1)** We leverage chemistry principles and propose the **TANimoto Group Overlap (TANGO)** reward function to generate molecules deemed synthesizable by retrosynthesis models with the presence of *enforced building blocks* using reinforcement learning (RL). **(2)** We show that generated molecules satisfy MPO objectives, and by design, enable the construction of *synthesis networks* where *common intermediates* branch towards diverse, high-reward molecules. **(3)** We show that letting a *general purpose* model *freely learn* (using incentives), can be a productive approach to optimizing challenging synthesizability objectives.

## 2. Related Work

**Retrosynthesis Models.** Retrosynthesis planning aims to find a set of commercial building blocks and viable chemical transformations that can be combined to synthesize a target molecule. Existing works encode chemically plausible transformations either as reaction templates (coded patterns) (Chen & Jung, 2021; Xie et al., 2023) or template-free approaches (learn from data) operating on SMILES strings (Liu et al., 2017; Segler & Waller, 2017; Schwaller et al., 2020; Thakkar et al., 2023; Han et al., 2024) or graphs (Sacha et al., 2021; Zhong et al., 2023). Subsequently, multi-step retrosynthesis planning is tackled by coupling a search algorithm such as Monte Carlo tree search (Segler et al., 2018), Retro* (Chen et al., 2020), Planning with Dual Value Networks (PDVN) (Liu et al., 2023), or the recent Double-Ended Synthesis Planning (DESP) (Yu et al., 2024). With retrosynthesis planning being a ubiquitous task in molecular discovery, many platform solutions exist, including SYNTHIA (Szymkuć et al., 2016; Grzybowski et al., 2018), AiZynthFinder (Genheden et al., 2020; Saigiridharan et al., 2024), ASKCOS (Coley et al., 2019; Tu et al., 2025), Eli Lilly's LillyMol (Watson et al., 2019), Molecule.one's M1

platform (Molecule.one), and IBM RXN (Schwaller et al., 2020). In the context of generative molecular design, retrosynthesis models are usually used for post-hoc filtering due to their inference cost, but recent work has shown that with a sample-efficient model, they can be incorporated directly as an optimization objective (Guo & Schwaller, 2024c).

**Synthesizability-constrained Molecular Generation.** Bridging concepts from retrosynthesis, synthesizability-constrained models anchor molecular generation by enforcing a set of valid chemical transformations (Vinkers et al., 2003; Hartenfeller et al., 2012; Ghiandoni et al., 2022; 2024; Bradshaw et al., 2019; 2020; Korovina et al., 2020; Gao et al., 2022; Seo et al., 2023; Koziarski et al., 2024; Gao et al., 2024; Cretu et al., 2024; Luo et al., 2024; Gottipati et al., 2020; Horwood & Noutahi, 2020; Fialková et al., 2021; Jocys et al., 2024; Seo et al., 2024). To date, there are no molecular generative models that can enforce the presence of specific building blocks in the synthesis graph and the closest works are SynNet (Gao et al., 2022) and the very recent SynFormer (Gao et al., 2024) models which can condition on a target molecule to propose a synthetic route. Current synthesizability-constrained approaches cannot reliably (or are sample-inefficient) satisfy MPO objectives which is a necessary requirement for practical applications. In this work, we show that a general-purpose model, can generate synthesizable molecules that satisfy MPO objectives while *enforcing* the presence of a small set of building blocks either at the start of the synthesis (starting-material constrained), as a common intermediate (intermediate-constrained), or non-commercial building blocks that diverge to diverse, favorable generated molecules (divergent synthesis) (Fig. 1). To our knowledge, there are only several works (Johnson et al., 1992; Yu et al., 2022; 2024; Szymkuć et al., 2016; Grzybowski et al., 2018) that enable some notion of building block-constrained synthesis planning. In particular, the very recent DESP (Yu et al., 2024) retrosynthesis search algorithm proposes a bidirectional search that can constrain on a starting-material. Our work differs in that we are not proposing a search algorithm, but rather the first *generative* approach that *jointly* tackles constrained synthesizability and MPO. Moreover, our framework can consider the constraint of *many* building blocks simultaneously.

## 3. Methods

In this section, we describe the problem formulation, the generative model, the **TANGO** reward function, and the experimental setup.

**Constrained Synthesizability Problem Formulation.** In synthesis planning, the goal is to propose a valid synthetic route to a target molecule using (commercially) available

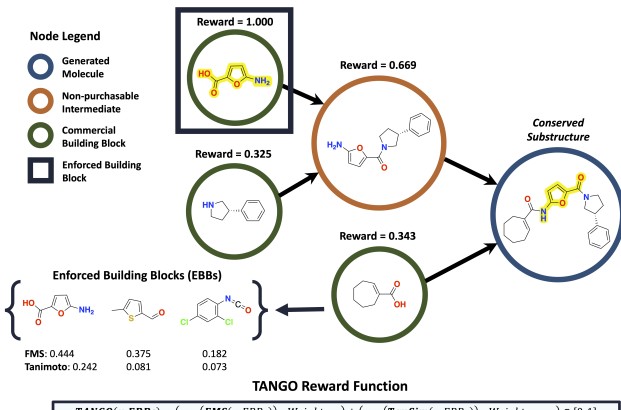

**Enforced Building Blocks (EBBs)**

FMS: 0.444     0.375     0.182
Tanimoto: 0.242    0.081    0.073

**TANGO Reward Function**

$$TANGO(x, EBBs) = (\max(FMS(x, EBBs)) * Weight_{FMS}) + (\max(TanSim(x, EBBs)) * Weight_{TanSim}) \in [0, 1]$$

*Figure 2.* TANGO reward function: the maximum *similarity* between *every* non-root node (generated molecule) molecule and the set of enforced building blocks. Every synthesizable generated molecule returns a non-zero reward.

building blocks, $B$, and a set of reaction rules, $R$. We define a *synthesis graph*, $G(M, R)$, where each node represents an intermediate molecule, $m$, that need not necessarily be an available building block, $b$, and the edges represent reactions, $r \in R$. The *depth* of a node is the number of edges from the root node (the target molecule). A valid synthetic route requires that all leaf nodes correspond to commercially available building blocks, $b \in B$. We further define *enforced* building blocks, $B_{enf} \subseteq B$. In practice, $|B_{enf}| << |B|$, and in this work, we consider $|B_{enf}| \in \{10, 100\}$. We address three cases of constrained synthesis in this work:

**Case 1: Starting-material Constrained Synthesis.** A synthesis graph is considered *starting-material constrained* if at least one leaf node, $m \in G(M, R)$, satisfies both of the following conditions: (1) $m = b \in B_{enf}$, and (2) depth($m$) = max depth:

$$\exists m \in G(M, R) \text{ s.t. depth}(m) = \max \text{ depth and } m = b \in B_{enf}$$

A practical reason why one would want to enforce a starting-material constraint is that they may be inexpensive reagents. As such, they can be obtained in larger quantities for use in multi-step synthesis, which necessarily loses material at every synthetic step.

**Case 2: Intermediate Constrained Synthesis.** A synthesis graph is considered *intermediate constrained* (general case of starting-material constrained) if at least one intermediate node, $m \in G(M, R)$, belongs to $B_{enf}$:

$$\exists m \in G(M, R) \text{ s.t. } m \in B_{enf}$$

**Case 3: Divergent Synthesis.** A synthesis graph is con-

sidered *divergent* if at least one intermediate node, $m \in G(M, R)$, satisfies both of the following conditions: (1) $m = b \in B_{enf}$, and (2) all $b \in B_{enf}$ are non-commercial. The nuance of *non-commercial* is that they can be highly specific building blocks and potentially much larger in size than common commercial building blocks, which can enable late-stage functionalization (Castellino et al., 2023):

$$\exists m \in G(M, R) \text{ s.t. } \forall b \in B_{enf}, b \text{ is non-commercial, and } m = b \in B_{enf}$$

**TANGO Reward Function.** In the context of generative molecular design, previous work has shown that retrosynthesis models can be treated as an oracle and directly optimized for (Guo & Schwaller, 2024c). The effect is that generated molecules are synthesizable, as deemed by retrosynthesis models (from here on, this will just be referred to as "synthesizable", for brevity). In that work, the authors adopted a brute-force approach to *learning* synthesizability, despite the reward signal being binary, i.e., $R \in \{0, 1\}$, denoting whether a molecule is synthesizable or not. This worked because there are enough molecules that are synthesizable, making the optimization landscape not *too* sparse. However, in the constrained synthesis setting, it is highly unlikely that a synthesizable molecule will also contain an enforced building block in its synthesis graph, especially when the number of $B_{enf}$ is small, which is common in real-world applications (Granda et al., 2018; Wołos et al., 2020; 2022). Consequently, this is a very sparse reward environment: without a way to inform the model if it is getting "closer" to achieving the goal, learning becomes extremely challenging. Drawing inspiration from RL, quantifying the degree to which an arbitrary goal is achieved, while intending for another goal is sometimes referred to as *hindsight* (Rauber et al., 2019). Intuitively, defining a reward signal that is not exactly the target objective but is *informative* to achieving the target objective should guide learning (Andrychowicz et al., 2017; Rauber et al., 2019), and can be done through *reward shaping* (Ng et al., 1999; Silver et al., 2021).

In this work, we propose the **TANimoto Group Overlap (TANGO)** reward function that leverages chemistry inductive bias to transform the *sparse* reward environment associated with constrained synthesis, to a *dense* reward environment. Specifically, for every synthesizable molecule, TANGO provides a signal on whether the model is "closer" to incorporating $B_{enf}$. Given $G(M, R)$, this is achieved by a notion of *similarity* between every node, $m$, and $B_{enf}$. We draw inspiration from previous works that leverage Tanimoto similarity (sub-graph similarity) for retrosynthesis (Coley et al., 2017; Zhang et al., 2024). However, Tanimoto similarity alone is insufficient as chemical *reactivity* is often associated with *functional groups* and their neighborhoods which dictate incompatibilities (Molga et al., 2019). Cor-

respondingly, we augment Tanimoto similarity with Functional Group (FG) Overlap (does a given molecule have similar functional groups to $B_{enf}$?) and Fuzzy Matching Substructure (FMS) (what is the maximum substructure overlap by atom count to $B_{enf}$?). We note that FMS, if enforcing exact atom hybridization, atom type, chirality, and whether the atom is part of a ring, also *implicitly* considers functional group overlap. In Appendix E, we systematically evaluated various TANGO formulations and their ability to distinguish "goodness" and conclude that equal weighting (0.5) of Tanimoto similarity (TS) and FMS yields the best learnable signal:

$$TANGO(m, B_{enf}) = TS(m, B_{enf}) \cdot 0.5 + FMS(m, B_{enf}) \cdot 0.5 \in [0, 1] \quad (1)$$

We note that since the maximum value of Tanimoto similarity and FMS is 1, TANGO is by design already normalized $\in [0, 1]$. The TANGO reward is the maximum value between all non-root nodes, $m \in G(M, R)$ (Algorithm 1). It follows that the *type* of constrained synthesizability can be controlled by a simple toggle of whether only nodes at max depth (*starting-material*) be considered or any node (*intermediate or divergent*).

---

**Algorithm 1** TANGO Reward Calculation

**Input:**
$G(M, R)$          ▷ Synthesis graph of generated molecule
$B_{enf}$          ▷ Enforced building blocks
*max depth*          ▷ Graph depth of terminal leaf nodes
*enforce start*          ▷ Boolean flag for starting-material constraint

**Output:** *reward*

$reward \leftarrow 0$
// Traverse all non-root nodes in the synthesis graph
**foreach** node $m \in G(M, R)$ **and** $depth(m) > 0$ **do**
     // Starting-material constrained or not
     **if** *enforce start* **then**
         **if** $depth(m) \neq max\ depth$ **then**
             **continue**
         **end**
     **end**
     $reward \leftarrow max(reward, \text{TANGONodeReward}(m, B_{enf}))$
**end**

**Function** TANGONodeReward($node$, $B_{enf}$):
     $node\_reward \leftarrow 0$
     // Loop through all enforced building blocks
     **foreach** $b_{enf} \in B_{enf}$ **do**
         // Compute current block's reward
         $TanSim \leftarrow \text{ComputeTanimotoSimilarity}(node, b_{enf})$
         $FMS \leftarrow \text{ComputeFMS}(node, b_{enf})$
         $block\_reward \leftarrow TanSim \cdot 0.50 + FMS \cdot 0.50$
         $node\_reward \leftarrow max(node\_reward, block\_reward)$
     **end**
     **return** $node\_reward$
**return** TANGO_reward

---

**Molecular Generative Model.** Here, we build on Saturn (Guo & Schwaller, 2024b) which is a general-purpose autoregressive language-based model operating on SMILES strings (Weininger, 1988). Saturn uses the Mamba (Gu & Dao, 2023) architecture and performs goal-directed generation using RL. The key mechanism is combining SMILES augmentation (Bjerrum, 2017) with experience replay (Lin, 1992) which directly controls the exploration-exploitation

trade-off. In the original work, the authors found that *aggressive* local sampling in chemical space improves sample efficiency across various drug discovery case studies. By contrast, we show that the constrained synthesizability setting necessitates a more exploratory behavior. We pre-train Saturn on PubChem (Kim et al., 2023) after data pre-processing (see Appendix A for details).

**Retrosynthesis Model.** In this work, we integrate Syntheseus (Maziarz et al., 2023), which is a wrapper around various retrosynthesis models and search algorithms, into Saturn. Through Syntheseus, we use MEGAN (graph-edits based) (Sacha et al., 2021) as the single-step retrosynthesis model coupled with the Retro* (Chen et al., 2020) search algorithm with default hyperparameters. MEGAN was chosen due to its fast inference speed but we emphasize that our framework is retrosynthesis model-agnostic.

**Commercial Building Blocks.** In this work, $B$ is comprised of the 'Fragment' and 'Reactive' subsets of ZINC (Sterling & Irwin, 2015) (17,721,980) which are part of the commercial building block stock used in the AiZynthFinder (Genheden et al., 2020; Saigiridharan et al., 2024) retrosynthesis model. We note that the size of $B$ is *much* larger than employed in previous synthesizability-constrained works (Gao et al., 2022; Luo et al., 2024; Gao et al., 2024; Seo et al., 2024) (which commonly use Enamine REAL (Grygorenko et al., 2020) US Stock: 223,244 molecules and recently Enamine Comprehensive Catalogue: 1,193,871 molecules). Therefore, an additional result in this paper is showing that our framework can navigate an *enormous* synthesizable space. We also want to highlight that it is straightforward to further increase the size of $B$, and does not require re-training of the generative model. Next, $B_{enf} \subset B$ is randomly sampled with the following criteria: 150 < molecular weight < 200, no aliphatic carbon chains longer than 3, exclude charged molecules, if rings are present, enforce size $\in \{5, 6\}$, and molecules must contain at least one nitrogen, oxygen, or sulfur atom. We believe this criteria is a reasonable representation of simple building blocks applicable to drug design (see Appendix B for more details) We consider $|B_{enf}| \in \{10, 100\}$ and denote these $B_{enf-10}$ and $B_{enf-100}$, respectively.

**Drug Discovery Case Study.** The MPO optimization task is to generate molecules with optimized QuickVina2-GPU-2.1 (Trott & Olson, 2010; Alhossary et al., 2015; Tang et al., 2023) docking scores against ATP-dependent Clp protease proteolytic subunit (ClpP) (Mabanglo et al., 2023) (implicated in cancer), high QED (Bickerton et al., 2012), and are synthesizable with either the *starting-material*, *intermediate*, or *divergent synthesis* constraints.

**Experimental Details.** For method development (see Appendix E for all experimental results), we ran every experiment across 5 seeds (0-4 inclusive) with varying oracle

budgets. Once we identified optimal hyperparameters, we ran all main result experiments across 10 seeds (0-9 inclusive) with a 10,000 oracle budget, and reported the wall time to promote practical application. As our framework is, to the best of our knowledge, the first *generative* approach that tackles constrained synthesizability, we focus our investigation on the optimization dynamics and implications of TANGO.

**Metrics.** We report **Non-solved** and **Solved (Enforced)** as the number of generated molecules that the retrosynthesis model deems unsynthesizable (no route returned) and is synthesizable *with* the presence of an enforced building block, respectively. Note that **Solved (Enforced)** is a much more challenging metric than *just* synthesizable, which previous work has shown is directly learnable (Guo & Schwaller, 2024c). We further report **N** as the number of replicates out of 10 seeds where **Solved (Enforced)** > 0, and the mean and standard deviation for the **# Unique Enforced Blocks**, denoting how many *unique* enforced building blocks are in the routes for the **Solved (Enforced)** molecules. Next, we pool all **Solved (Enforced)** molecules and report the mean and standard deviation of the **# Reaction Steps**. Similarly, we report the mean and standard deviation of docking scores and QED values across varying intervals. Jointly optimizing for constrained synthesizability, minimizing docking scores, and maximizing QED values is the MPO objective and a robust model should be able to achieve this.

## 4. Results and Discussion

### 4.1. Making Constrained Synthesizability *Learnable*

**Understanding the Optimization Dynamics.** In Appendix E, we performed extensive ablation studies to understand the optimization dynamics that enable direct optimization of constrained synthesizability. We summarize our observations and show key results in Table 1: firstly, TANGO is the most consistent *learnable* reward function that also enables MPO. While just Tanimoto similarity as the reward function can lead to successful runs, it is less stable (seeds can fail and much higher variance) and MPO is considerably worse, as docking and QED is optimized to a much lesser extent than TANGO. FMS as a reward function is also successful, but generates very few constrained synthesizable molecules. Therefore, through ablation studies, we show that *it takes two to tango* because TANGO's performance is much more consistent and *robust* (across seeds) than using just Tanimoto similarity or FMS alone. Thus far, TANGO was formulated with equal weighting to FMS and Tanimoto similarity. We next investigated varying the weighting, assigning 0.75 to either FMS or Tanimoto Similarity and 0.25 to the other. The results in Appendix E.5 show that putting more weight on Tanimoto similarity leads to more constrained synthesizable molecules, but at the ex-

pense of worse MPO. Since MPO is vital for practicality, we chose to designate TANGO with equal weighting the default reward function. Next, we found that the default hyperparameters of Saturn (Guo & Schwaller, 2024b) are *too exploitative* and disadvantageous in this optimization setting. Relaxing this behavior makes constrained synthesizability much more *consistently* learnable (under the fixed oracle budget). Lastly, once molecules with a specific enforced building block are generated, Saturn heavily focuses on that building block, such that within the same generative experiment, often only one *unique* block is enforced (but variable across seeds). We do not consider this a disadvantage as it enables the construction of *divergent synthesis networks* where a common block branches towards many optimal molecules. We discuss the implications of this behavior from a generative perspective in the next section.

**Constrained Synthesizability Results.** Table 2 shows the results with $B_{enf-10}$ and $B_{enf-100}$ using TANGO (equal weighting). We make the following observations: firstly, all constraints can be learned within the 10,000 oracle budget (approximate 8.5 hours wall time). Secondly, all runs generate non-solvable molecules and many solvable molecules do not contain the enforced blocks, as expected (see **Non-solved** and **Solved (Enforced)**). Nonetheless, generated molecules can achieve docking scores < -10 (considered optimal in previous works and is better than the reference ligand (Koziarski et al., 2024; Guo & Schwaller, 2024c)) and optimal QED values. This demonstrates the capability to perform MPO while also optimizing for constrained synthesizability. Thirdly, **# Unique Enforced Blocks** is relatively low as we observed that once the model incorporates *one* enforced building block, it focuses on generating molecules whose syntheses can be decomposed to that specific block, since the reward it obtains is high and there is a degree of exploitation. Fourthly, the starting-material constraint is more difficult for $B_{enf-100}$ but unexpectedly, not for $B_{enf-10}$. We speculate the reason for this is exactly due to exploitation behavior. Since TANGO returns the max reward in the synthesis tree (comparing to all $B_{enf}$), it is possible that more blocks can be a hindrance when there are specific blocks that are particularly favorable. We emphasize that across different seeds, the enforced building blocks can be different, which is important as one could run multiple experiments in parallel and pool the results. Fifthly, we ran the same experiments *without* the QED objective and the optimization task becomes easier (as expected), with higher **Solved (Enforced)** and molecules with docking scores < -10 (Appendix E.6). We ran these sets of experiments for completeness and comparison only, as particularly low QED can result in lipophilic molecules that can be promiscuous binders (Arnott & Planey, 2012). We highlight that the **# Reaction Steps** is generally short, which shows that optimizing for constrained synthesizability does not lead to

*Table 1.* Results for the promising reward functions: **Tanimoto Similarity (TanSim)**, **Fuzzy Matching Substructure (FMS)**, and **TANGO (equal weighting)**. All experiments were with 100 enforced building blocks. The mean and standard deviation across 5 seeds (0-4 inclusive) are reported. The number of replicates (out of 5) with at least 1 generated molecule that is synthesizable with an enforced building block is reported with **N**. The number of molecules (pooled across all successful replicates) are partitioned into different docking score thresholds and statistics reported. # Reaction Steps is also reported for the pooled generated molecules that have an enforced block. The total number of molecules in each pool across the 5 seeds is denoted by **M**. For the docking score intervals, we report the scores and QED values.

| Configuration | Synthesizability | | Docking Score Intervals (QED Annotated) | | |
|---|---|---|---|---|---|
| | Non-solved | Solved (Enforced) | DS < -10 | -10 < DS < -9 | -9 < DS < -8 |
| TanSim | $2275 \pm 143$ | $1863 \pm 1827$ | $-10.36 \pm 0.24$ (M=30) | $-9.36 \pm 0.23$ (M=487) | $-8.43 \pm 0.25$ (M=2389) |
| (N=3) | | | $0.72 \pm 0.09$ | $0.76 \pm 0.10$ | $0.79 \pm 0.09$ |
| FMS | $1693 \pm 174$ | $114 \pm 205$ | $-10.36 \pm 0.40$ (M=5) | $-9.41 \pm 0.25$ (M=53) | $-8.48 \pm 0.25$ (M=173) |
| (N=5) | | | $0.75 \pm 0.15$ | $0.85 \pm 0.09$ | $0.85 \pm 0.07$ |
| TANGO | $2229 \pm 325$ | $1743 \pm 715$ | $-10.34 \pm 0.25$ (M=218) | $-9.44 \pm 0.25$ (M=1606) | $-8.49 \pm 0.26$ (M=2206) |
| (N=5) | | | $0.77 \pm 0.11$ | $0.83 \pm 0.10$ | $0.82 \pm 0.10$ |

| Configuration | # Reaction Steps | # Unique Enforced Blocks | Oracle Budget (Wall Time) |
|---|---|---|---|
| TanSim | $2.18 \pm 1.16$ (M=9319) | $1.33 \pm 0.47$ | 10,000 (9h 11m $\pm$ 48m) |
| FMS | $1.78 \pm 1.04$ (M=570) | $1.8 \pm 0.75$ | 10,000 (7h 50 $\pm$ 26) |
| TANGO | $2.35 \pm 1.24$ (M=8719) | $2.2 \pm 0.75$ | 10,000 (8h 12m $\pm$ 15m) |

inefficient synthesis plans. Moreover, amongst the most recent synthesizability-constrained models (Seo et al., 2024), our framework outputs shorter synthesis routes, on average, despite operating in the much more challenging setting. This is because our framework can perform MPO and optimizing for QED implicitly yields shorter synthesis routes, on average, as it constrains the molecular weight. We note that the **# Reaction Steps** in the divergent synthesis results are longer because it takes one step in the first place, to arrive at the divergent blocks. Finally, all runs only took on average, 8.5 hours on a single GPU, which is reasonable, as many commercial drug discovery projects run their generative experiments for 24-72 hours (Livne et al., 2024).

**Are the Results by Chance?** While the results thus far were promising, we noticed that many runs (across seeds) converged to the same three enforced building blocks (Fig. 3). We questioned whether the success was simply due to these "lucky" blocks. Therefore, we performed a set of ablation experiments by removing these blocks and re-running all configurations in Table 2. The results show that the model can generate optimal molecules with other enforced building blocks (Appendix E.7). These runs were much less successful (across seeds) but recovered when removing the QED objective. This suggests that the model learns to use certain blocks that are more aligned with the objective function. Next, we further push our framework with an enforced building block set of *5 molecules* dissimilar to the "lucky" blocks. While much more challenging (most runs are unsuccessful under the strict oracle budget), this is still possible, with the routes containing Suzuki coupling reactions (see Appendix E.8 and Fig. 12), which is notably different to the amide coupling reactions in Fig. 3.

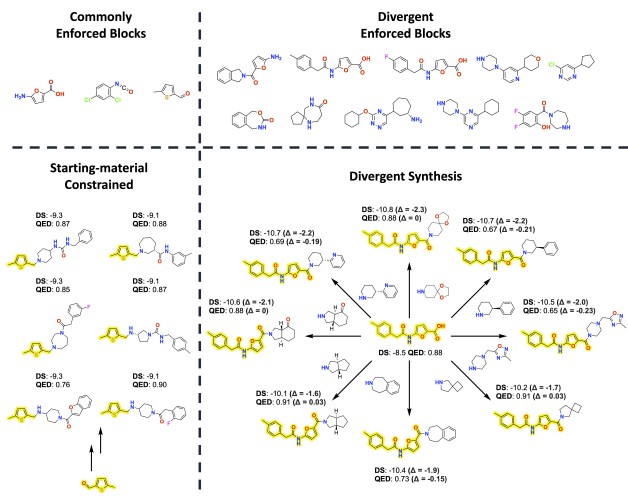

*Figure 3.* Example generated molecules under the **starting-material** and **divergent synthesis** (one-step synthesis from a non-commercial common intermediate to diverse, high-reward molecules) constraints. The docking scores and QED values are annotated. For the divergent synthesis graph, the $\Delta$ docking score (negative is better) and QED (positive is better) are additionally annotated.

**Synthesis Networks.** Next, we tackle *divergent synthesis* by incorporating larger, non-commercial molecules in the enforced building blocks set. We curated a set of 10 non-commercial blocks from solved synthesis routes (Fig. 3) and ran the same experimental set-up. Table 2 shows that this is also learnable within the oracle budget, albeit much less consistently as only 4/10 seeds were successful. We still argue that our framework is robust as this

*Table 2.* Constrained synthesizability results. The reward function is TANGO (equal FMS and Tanimoto similarity weighting). "SM" denotes starting-material constrained. "Unconstrained" denotes the experiment without enforcing building blocks, as a comparison. The mean and standard deviation across 10 seeds (0-9 inclusive) are reported. The number of replicates (out of 10) with at least 1 generated molecule that is synthesizable with an enforced building block is reported with **N**. The number of molecules (pooled across all successful replicates) are partitioned into different docking score thresholds and statistics reported. # Reaction Steps is also reported for the pooled generated molecules that have an enforced block. The total number of molecules in each pool across the 10 seeds is denoted by **M**. For the docking score intervals, we report the scores and QED values.
[a] Denotes how many molecule are solvable by the retrosynthesis model. There is no notion of *enforced* in the unconstrained setting.

| Configuration | Synthesizability | | Docking Score Intervals (QED Annotated) | | |
|---|---|---|---|---|---|
| | Non-solved | Solved (Enforced) | DS < -10 | -10 < DS < -9 | -9 < DS < -8 |
| 100 Blocks | 2288 ± 305 | 2111 ± 1169 | -10.36 ± 0.28 (M=487) | -9.42 ± 0.24 (M=3096) | -8.47 ± 0.26 (M=5904) |
| (N=10) | | | 0.79 ± 0.09 | 0.82 ± 0.09 | 0.81 ± 0.09 |
| 100 Blocks (SM) | 1879 ± 186 | 1524 ± 502 | -10.41 ± 0.31 (M=120) | -9.41 ± 0.24 (M=985) | -8.43 ± 0.25 (M=3156) |
| (N=10) | | | 0.78 ± 0.09 | 0.81 ± 0.09 | 0.81 ± 0.09 |
| 10 Blocks | 2425 ± 288 | 984 ± 1181 | -10.38 ± 0.30 (M=659) | -9.46 ± 0.25 (M=3981) | -8.57 ± 0.25 (M=2419) |
| (N=6) | | | 0.79 ± 0.10 | 0.83 ± 0.09 | 0.83 ± 0.10 |
| 10 Blocks (SM) | 2228 ± 182 | 1004 ± 925 | -10.37 ± 0.27 (794) | -9.46 ± 0.24 (M=3881) | -8.54 ± 0.25 (M=2790) |
| (N=9) | | | 0.80 ± 0.09 | 0.83 ± 0.09 | 0.84 ± 0.10 |
| Divergent Blocks | 2166 ± 202 | 651 ± 1238 | -10.36 ± 0.26 (M=187) | -9.41 ± 0.24 (M=1311) | -8.48 ± 0.25 (M=2694) |
| (N=4) | | | 0.84 ± 0.10 | 0.86 ± 0.07 | 0.86 ± 0.07 |

| Configuration | Synthesizability | | Docking Score Intervals (QED Annotated) | | |
|---|---|---|---|---|---|
| | Non-solved | Solved[a] | DS < -10 | -10 < DS < -9 | -9 < DS < -8 |
| Unconstrained | 1827 ± 191 | 8127 ± 196 | -10.36 ± 0.28 (M=5489) | -9.42 ± 0.24 (M=20099) | -8.47 ± 0.26 (M=26710) |
| (N=10) | | | 0.87 ± 0.07 | 0.88 ± 0.07 | 0.87 ± 0.08 |

| Configuration | # Reaction Steps | # Unique Enforced Blocks | Oracle Budget (Wall Time) |
|---|---|---|---|
| 100 Blocks | 2.37 ± 1.27 (M=21115) | 2 ± 0.63 | 10,000 (8h 31m ± 40m) |
| 100 Blocks (SM) | 1.49 ± 0.91 (M=15247) | 1.9 ± 0.7 | 10,000 (8h 33m ± 30m) |
| 10 Blocks | 2.70 ± 1.20 (M=9845) | 1 ± 0 | 10,000 (8h 29m ± 30m) |
| 10 Blocks (SM) | 2.59 ± 1.04 (M=10040) | 1 ± 0 | 10,000 (8h 39m ± 24m) |
| Divergent Blocks | 3.68 ± 1.08 (M=6512) | 1.75 ± 0.83 | 10,000 (8h 52m ± 42m) |

| Configuration | # Reaction Steps | # Unique Enforced Blocks | Oracle Budget (Wall Time) |
|---|---|---|---|
| Unconstrained | 1.86 ± 1.19 (M=81829) | N/A | 10,000 (5h 34m ± 39m) |

is a much more challenging task (one can also increase the oracle budget which leads to more successful seeds, as we show in Appendix E.9) and we wanted to show the model can *learn* to enforce these large building blocks *from scratch*. This opens up practical applications for late-stage functionalization, commonly employed in drug discovery (Castellino et al., 2023). Correspondingly, Fig. 3 shows example synthesis networks using results from the *starting-material* and *divergent synthesis* constrained experiments. All generated molecules achieve optimal docking scores (although starting-material constrained resulted in slightly worse scores) and QED values. In the divergent synthesis case, a one-step amide coupling reaction from the enforced block leads to notably improved docking scores, though sometimes with lower QED. Examples of full synthesis routes are shown in Appendix F.

## 4.2. Learning a Desirable Distribution

Fundamentally, generative models learn to model distributions. In this section, we further demonstrate that TANGO is a *learnable* reward function and that the modeled distribution shifts to satisfy the MPO objective. To do so, we take each final model checkpoint (across the 10 seeds) from the experiment in Table 2 with 100 enforced building blocks (and with QED) and sample 1,000 unique molecules. Fig. 4a shows that a considerable number of sampled molecules are *jointly* synthesizable with an enforced building block (**Solved (Enforced)**). The distribution shift is apparent when compared to 1,000 unique molecules sampled from the *pre-trained model* (before RL), which mostly generates unsynthesizable molecules (**Non-solved**). Fig 4b pools (across the 10 seeds) all the **Solved (Enforced)** molecules and shows the density of docking and QED scores which have shifted towards favorable values. Next, we take the sampled molecules from one seed and plot a UMAP (McInnes et al.,

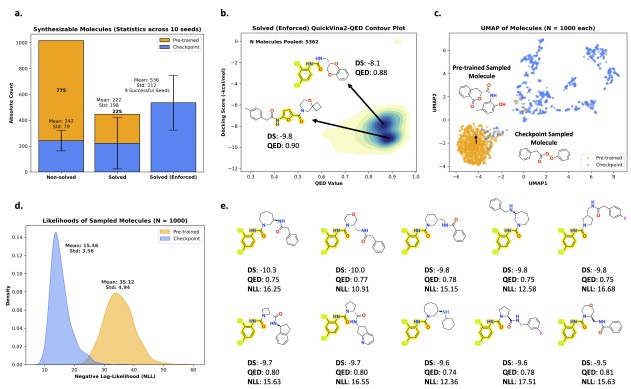

*Figure 4.* The model learns a distribution of molecules that satisfy the MPO objective. The final model checkpoint from the 100 enforced building blocks experiment (*all 10 seeds*) was used to sample 1,000 *unique* molecules. **a.** Counts of solvable molecules from the checkpoints with the mean and standard deviation reported (non-bolded). 1/10 final model checkpoints was unable to yield "Solved (enforced)" molecules. The pre-trained model (before RL) generates mostly unsynthesizable molecules and no synthesizable molecules with enforced blocks (**metrics are bolded**). **b.** Docking Score (DS) and QED values of the pooled Solved (Enforced) molecules across all seeds. *c, d, e uses 1,000 unique molecules sampled from one final model checkpoint.* **c.** UMAP of sampled molecules compared to the pre-trained model. **d.** Negative log-likelihoods (NLLs) of the sampled molecules. It is much more likely to generate the sampled molecules under the final model checkpoint. **e.** Top-10 (by docking score) molecules with the enforced building block highlighted. The NLLs are similar.

2018) embedding comparing to the molecules sampled from the pre-trained model. It is clear that the checkpoint sampled molecules are dissimilar but we show that the learned distribution is not *perfect*, as the final checkpoint still sometimes samples ill-suited (based on the MPO objective) molecules that are similar to the pre-trained model. Subsequently, we take the sampled molecules from the final model checkpoint and compare the negative log-likelihoods as measured by this checkpoint and the pre-trained model. We make two observations: firstly, the molecules are much more likely under the checkpoint, unsurprisingly. But secondly, and more importantly, the likelihoods from the checkpoint puts more probability mass in a narrower region. We now cross-reference Fig. 4e which shows the top-10 sampled molecules (by docking score) which all share the same enforced building block. The likelihoods are not drastically different, and shows that some exploitation during RL is advantageous as the likelihoods of molecules which share a common structure can be quite similar. Very specifically, given a favorable molecule represented as a SMILES, Saturn's (Guo & Schwaller, 2024b;a) mechanism of optimization involves making it likely to generate *any* SMILES form of the same molecular graph. If it is likely to generate *any* SMILES sequences of the same favorable molecule,

small changes to the generated sequence amounts to small chemical changes, which can be advantageous, as similar molecules, on average, have similar properties. The model learns to use the building blocks in a way that performs local exploration and assigns a relatively similar likelihood to the neighborhood of molecules. Overall, the results show that taking a *general-purpose* model and *incentivizing* the learning process with TANGO, can shift the modeled distribution to one that captures constrained synthesizability while simultaneously satisfying MPO objectives. This is practically useful, as one could simply sample molecules from model checkpoints to get more desirable molecules (5 seconds to sample 1,000 unique molecules).

## 5. Conclusion

In this work, we proposed a novel reward function called **TANimoto Group Overlap (TANGO)** that can guide a general-purpose molecular generative model to *directly* optimize for constrained synthesizability while also simultaneously performing MPO. This work is the first example of a *generative* approach for constrained synthesizability, and tackles various degrees of constraints that are practically important in real-world applications: starting-material, intermediate, and divergent synthesis constraints (Fig. 3). The results show that the generative model, Saturn (Guo & Schwaller, 2024b), when augmented with TANGO, can generate optimal molecules for a drug discovery case study involving molecular docking (Table 2). Moreover, the results show that our framework can learn to enforce building block sets as small as 10 and even 5 (Appendix E.8), which is practically relevant for re-purposing building blocks into useful molecules (Granda et al., 2018; Wołos et al., 2020; 2022). From a generative model perspective, we have shown that optimizing for constrained synthesizability necessitates a better exploration-exploitation trade-off, providing practical insights into MPO in these settings. Furthermore, our results show that *incentivizing* an *unconstrained* model can lead to productive learning even in challenging synthesizability MPO settings. Our results clearly show that TANGO guides Saturn to learn a desirable distribution, as sampling molecules from the final model checkpoints yield molecules tailored to the MPO objective (Fig. 4). Our framework is general and can also be applied to the generative design of functional materials. Finally, "true synthesizability" depends on the accuracy of the retrosynthesis model and they are not perfect. It is likely that some routes generated are not synthetically feasible and/or lack regio- or stereo-selectivity (Molga et al., 2019). This is a limitation of current retrosynthesis models and is an ongoing challenge for improved synthesis planning. As we are not proposing a new retrosynthesis model, this is beyond the scope of this work, but we believe this is important to explicitly acknowledge.

## Impact Statement

This paper presents a generative method to design molecules that are synthesizable with pre-defined chemical reagents. As it is a general method, the model could be used for the design of potential therapeutics and functional materials. With proper experimental validation, designed molecules could have positive societal benefits such as treating diseases and mitigating climate change.

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

## A. PubChem Pre-processing and Saturn Pre-training

This section contains the full data pre-processing and pre-training pipeline starting from the raw PubChem which was downloaded from `https://ftp.ncbi.nlm.nih.gov/pubchem/Compound/Extras/`. The exact file is "CID-SMILES.gz".

The exact pre-processing steps along with the SMILES remaining after each step are:

1. Raw PubChem - 118,563,810

2. De-duplication - 118,469,904

3. Standardization (charge and isotope handling) based on `https://github.com/MolecularAI/ReinventCommunity/blob/master/notebooks/Data_Preparation.ipynb`. All SMILES that could not be parsed by RDKit were removed - 109,128,315

4. Tokenize all SMILES based on REINVENT's tokenizer: `https://github.com/MolecularAI/reinvent-models/blob/main/reinvent_models/reinvent_core/models/vocabulary.py`

5. Keep SMILES $\leq 80$ tokens, $150 \leq$ molecular weight $\leq 650$, number of heavy atoms $\leq 40$, number of rings $\leq 8$, Size of largest ring $\leq 8$, longest aliphatic carbon chain $\leq 4$ - 97,667,549

6. Removed SMILES containing the following tokens (due to undesired chemistry, low token frequency, and redundancy): [Br+2], [Br+3], [Br+], [C+], [C-], [CH+], [CH-], [CH2+], [CH2-], [CH2], [CH], [C], [Cl+2], [Cl+3], [Cl+], [ClH+2], [ClH2+2], [ClH3+3], [N-], [N@+], [N@@+], [NH+], [NH-], [NH2+], [NH3+], [NH], [N], [O+], [OH+], [OH2+], [O], [S+], [S-], [S@+], [S@@+], [S@@], [S@], [SH+], [SH-], [SH2], [SH4], [SH], [S], [c+], [c-], [cH+], [cH-], [c], [n+], [n-], [nH+], [nH], [o+], [s+], [sH+], [sH-], [sH2], [sH4], [sH], [s] - **88,618,780**

The final vocabulary contained 35 tokens (2 extra tokens were added, indicating <START> and <END>) and carbon stereochemistry tokens were kept. Saturn (Guo & Schwaller, 2024b) uses the Mamba (Gu & Dao, 2023) architecture and we used the default hyperparameters in the code-base. With the vocabulary size of 35, the model has 5,265,408 parameters. Saturn was pre-trained for 5 steps, with each step consisting of a full pass through the dataset. The model was pre-trained on a workstation with an NVIDIA RTX 3090 GPU and AMD Ryzen 9 5900X 12-Core CPU. The pre-training parameters were:

1. Training steps = 5

2. Seed = 0

3. Batch size = 512

4. Learning rate = 0.0001

5. Randomize (Bjerrum, 2017) every batch of SMILES

Relevant metrics of the pre-trained model (final model checkpoint) are:

1. Average negative log-likelihood (NLL) = 30.914

2. Validity (10k) = 98.74%

3. Uniqueness (10k) = 98.73%

4. Wall time = 106 hours (takes a relatively long time, though we only used 1 GPU for training. Pre-training also only needs to be done once.)

## B. Retrosynthesis Details

This section contains details on the retrosynthesis model, the commercial building blocks, and the enforced building blocks.

### B.1. Retrosynthesis Framework

In this work, we use Syntheseus (Maziarz et al., 2023) (benchmark platform and wrapper around retrosynthesis models and search algorithms) to run retrosynthesis. We integrate Syntheseus into Saturn (Guo & Schwaller, 2024b) and run the MEGAN (Sacha et al., 2021) single-step model with the Retro* (Chen et al., 2020) search algorithm. In the Syntheseus work, the authors standardize and benchmark many retrosynthesis models and configurations, reporting the inference time and accuracy (across various metrics). We chose MEGAN because it has the fastest inference time, although the top-k accuracies were lower than other models. We note that top-k single-step accuracy does not necessarily equate to better performance on multi-step retrosynthesis. Faster inference time allowed us to iterate experiments and hypotheses faster and is the main reason we chose MEGAN. Our framework is model-agnostic and any retrosynthesis model could be used in place of MEGAN. All MEGAN hyperparameters were tuned by the Syntheseus authors and we use them as is.

### B.2. Commercial Building Blocks

All retrosynthesis models require commercial building blocks, $B$. In this work, we use the 'Fragment' and 'Reactive' sub-sets of ZINC (Sterling & Irwin, 2015), equating to 17,721,980 building blocks. These sub-sets were obtained from the commercial building block stock used in AiZynthFinder (Genheden et al., 2020; Saigiridharan et al., 2024). Next, we consider two sets of enforced building blocks, $B_{enf-10} \subset B_{enf-100} \subset B$. The enforced building block sets (10 or 100) are sub-sets of $B$ and were randomly sampled following the criteria:

1. 150 < molecular weight < 200

2. No aliphatic carbon chains longer than 3

3. Exclude charged building blocks

4. If rings are present, enforce size $\in \{5, 6\}$

5. All building blocks must contain at least one nitrogen, oxygen, or sulfur atom

The criteria we defined are based on enforcing building blocks that are "simple, common, and relevant for drug-like molecules". While there is an inherent bias here, we emphasize that our TANGO framework is general and the set of enforced building blocks can be freely changed. Finally, we want to highlight an important implication when considering the commercial building blocks, $B$, and the generative model. Due to intentional data pre-processing of PubChem, which was used to pre-train Saturn, the generative model cannot generate all the atom types present in $B$. The specific atom types are phosphorus and silicon. We removed these atoms due to their seldom presence in "drug-like" molecules (although phosphorus is common in pro-drugs). The effect of this is that *some* commercial building blocks are not relevant, but we did not purge these and used the ZINC sub-sets as is. Similar to the enforced building blocks set, the set of commercial building blocks can also freely be changed. The sets of enforced building blocks are provided in the code-base.

## C. Compute Details

Every experiment (except pre-training Saturn) was run on a cluster equipped with NVIDIA L40S GPUs. As we used a SLURM queuing system, many jobs could be allocated the same GPU to run simultaneously. This makes the wall time for each individual run slower, but the total time to finish experiments is faster. We report the wall times as is.

## D. Docking Reward Shaping

Saturn expected every property to be optimized to have a normalized reward $\in [0, 1]$. TANGO and QED are already by design normalized but QuickVina2-GPU docking needs to be reward shaped. This is done by the shaping function shown here.

## E. TANGO Development and Ablations

In this section, we present the systematic development of TANGO, all ablation studies, and additional results. The section will be divided sequentially into sub-sections detailing our hypotheses, the experiments we ran to study them, and the

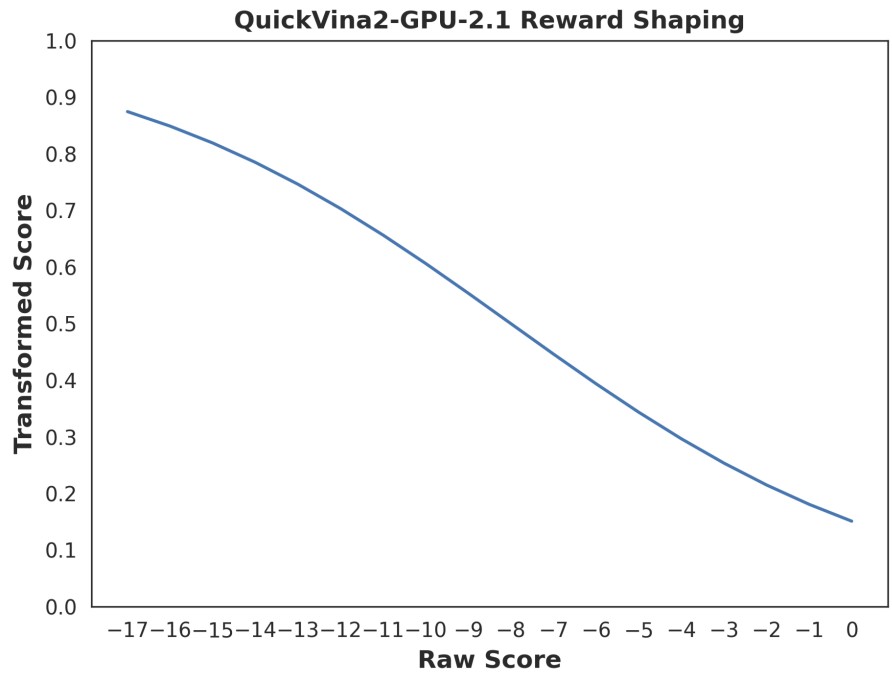

*Figure 5.* Reward shaping function for docking.

observations we made. All development experiments were run across 5 seeds (0-4 inclusive) while main result experiments were run across 10 seeds (0-9 inclusive). This information will be noted. The MPO objective is:

1. **Minimize QuickVina2-GPU-2.1** (Trott & Olson, 2010; Alhossary et al., 2015; Tang et al., 2023) docking scores against ATP-dependent Clp protease proteolytic subunit (ClpP) (Mabanglo et al., 2023) (implicated in cancer)

2. **Maximize QED** (Bickerton et al., 2012)

3. **Constrained Synthesizable**, as deemed by the MEGAN (Sacha et al., 2021) retrosynthesis model coupled with Retro* search (Chen et al., 2020)

Next, throughout TANGO development, we change the hyperparameters of Saturn, which directly control for the exploration-exploitation trade-off. We briefly summarize the key hyperparameters and their effect:

1. **Batch Size:** Lower is more exploitative

2. **Augmentation Rounds:** Higher is more exploitative

Finally, for all sets of experiments, we report metrics averaged across either 5 (0-4 inclusive) or 10 (0-9 inclusive) seeds. The number of seeds will be explicitly noted. The metrics are:

1. **Non-solved:** Number of generated molecules that do not have a solved synthetic route

2. **Solved:** Number of generated molecules that have a synthetic route **with at least 1 enforced building block**

3. **Docking Scores - QED:** Average and standard deviation of docking scores and QED values across various docking score thresholds. The rationale for this is because we want to optimize *all* objectives and analyzing different partitions is more informative

4. **Oracle Budget:** Number of oracle calls permitted

5. **Wall Time:** Compute time for the run

**Constrained Synthesizability** denotes either *start-material constrained* (enforced building blocks appearing at the max depth nodes in the synthesis graph), *intermediate-constrained* (enforced building blocks appearing anywhere in the synthesis graph), or *divergent synthesis* (enforced **non-commercial** building blocks appearing anywhere in the synthesis graph). This information will be explicitly noted. Finally, for brevity, we will write "synthesizable" to mean synthesizable, as deemed by the MEGAN retrosynthesis model.

### E.1. How can Constrained Synthesizability be made Learnable?

The starting point of TANGO development drew inspiration from (Coley et al., 2017; Zhang et al., 2024) which used Tanimoto similarity for retrosynthesis problems. We hypothesized that Tanimoto similarity is insufficient to inform chemical *reactivity*. Therefore, very initial experiments tried to "filter" nodes by matching for functional groups. Specifically, for every molecule generated that was synthesizable, there is a corresponding synthesis graph whose nodes are every intermediate molecule. The very first reward function traverses these nodes and computes the max Tanimoto similarity to the set of enforced building blocks, *provided that the node overlaps 75% of the functional groups with at least one of the enforced building blocks*, and returns this as the reward. With this initial reward formulation, we used Saturn's (Guo & Schwaller, 2024b) default hyperparameters of batch size 16 and 10 augmentation rounds. These parameters make the model perform local sampling in chemical space *aggressively*. We had run this experiment across 5 seeds (0-4 inclusive) with an oracle budget of 3,000 and only one seed was successful in generating *some* synthesizable molecules with the enforced building blocks. All seeds showed some *learning*, in that the average Tanimoto similarity of the synthesis graphs to the enforced building blocks was increasing (though it always stagnated). At the time, this was highly *irreproducible*, considering only 1/5 runs were successful. However, these failed runs gave us sets of molecules possessing various Tanimoto similarity to the enforced building blocks which we used to investigate various reward shaping functions. Specifically, we took the set of all generated molecules from one of the seeds and partitioned all the molecules that were synthesizable into the following Tanimoto similarity thresholds (to the enforced building blocks):

1. **Low:** $0.0 <$ TanSim $< 0.2$ (N = 237)

2. **Med:** $0.2 <=$ TanSim $< 0.3$ (N = 438)

3. **Med-High:** $0.3 <=$ TanSim $< 0.4$ (N = 734)

4. **High:** $0.4 <=$ TanSim $< 0.5$ (N = 38)

5. **Very-High:** $0.5 <=$ TanSim $> 1.0$ (N = 712)

The **reward distributions** of these sets of molecules were visualized under different **reward formulations** (Fig. 6). All comparison are to the set of enforced building blocks:

1. **Functional Groups (FG):** Mean or max functional groups overlap

2. **Tanimoto Similarity (TanSim):** Mean or max Tanimoto Similarity

3. **Fuzzy Matching Substructure (FMS):** Mean or max fraction of atoms in the maximum matching substructure

4. **TANGO-FG:** Max TanSim + Mean FG

5. **TANGO-FMS:** Max TanSim + Max FMS

6. **TANGO-All:** Max TanSim + Mean FG + Max FMS

Based on Fig 6, **Max TanSim**, **Max FMS**, and **TANGO-FMS** are able to separate the partitioned Tanimoto similarity intervals the best. These reward formulations are promising because they can distinguish between "closeness" to incorporating the enforced building blocks and enables a gradient for learning. It is important to know that this analysis has an explicit bias: we are assuming that Tanimoto similarity does in fact equate to being "closer", since we partitioned the generated set based on this. However, this gave us the first hypotheses to work with.

**Hypotheses:**

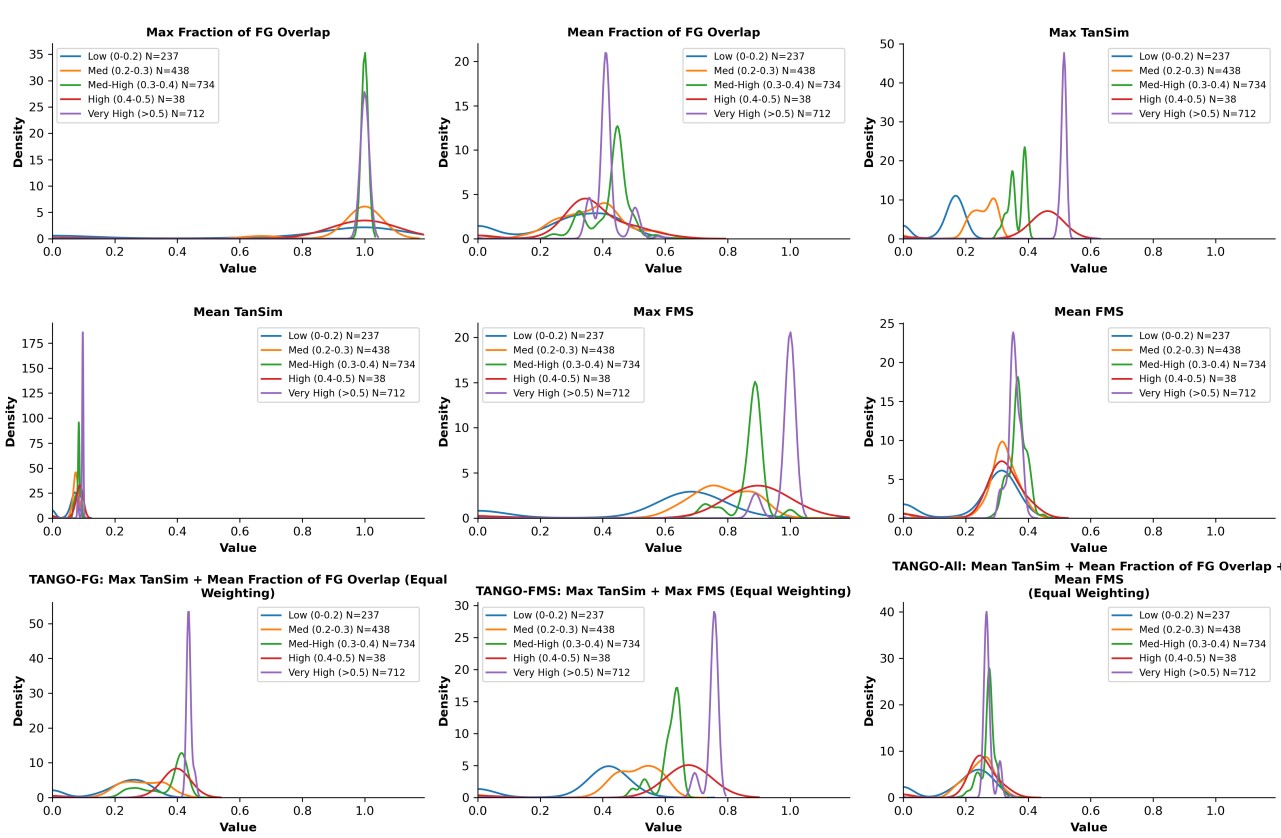

*Figure 6.* Reward distributions of different reward function formulations.

1. The initial run with 1/5 successful seeds used Batch Size = 16 and Augmentation Rounds = 10. This is likely too *exploitative*. Try a more *exploratory* sampling behavior with **Batch Size = 32 and Augmentation Rounds = 5**.

2. Try the most promising reward functions: **Max TanSim**, **Max FMS**, and **TANGO-FMS**.

**Fixed Parameters:**

1. **Oracle Budget** = 3,000

2. **Batch Size** = 32

3. **Augmentation Rounds** = 5

4. **Enforced Building Blocks** = 100

**Observations:** Table 3 shows the results with the mean and standard deviation across 5 seeds (0-4 inclusive). We make the following observations:

1. All reward functions can yield successful runs.

2. FMS and TanSim are inconsistent with 3/5 runs unsuccessful.

3. FMS finds very few molecules satisfying constrained synthesizability

4. TANGO-FMS yields the best average performance.

*Table 3.* Results for Section 1: How can Constrained Synthesizability be made Learnable? The mean and standard deviation across 5 seeds (0-4 inclusive) are reported. The number of replicates (out of 5) with at least 1 generated molecule that is synthesizable with an enforced building block is reported with **N**. The number of molecules (pooled across all successful replicates) are partitioned into different docking score thresholds and statistics reported. # Reaction Steps is also reported for the pooled generated molecules that have an enforced block. The total number of molecules in each pool across the 5 seeds is denoted by **M**. For the docking score intervals, we report the scores and QED values.

| Configuration | Synthesizability | | Docking Score Intervals (QED Annotated) | | |
|---|---|---|---|---|---|
| | Non-solved | Solved (Enforced) | DS < -10 | -10 < DS < -9 | -9 < DS < -8 |
| TanSim | 577 ± 32 | 333 ± 408 | None | -9.35 ± 0.24 (M=20) | -8.38 ± 0.25 (M=248) |
| (N=2) | | | N/A | 0.70 ± 0.08 | 0.73 ± 0.10 |
| FMS | 578 ± 30 | 9 ± 12 | None | None | -8.36 ± 0.26 (M=20) |
| (N=2) | | | N/A | N/A | 0.85 ± 0.04 |
| TANGO-FMS | 643 ± 23 | 476 ± 377 | -10.36 ± 0.25 (M=10) | -9.40 ± 0.24 (M=146) | -8.42 ± 0.25 (M=596) |
| (N=5) | | | 0.70 ± 0.03 | 0.70 ± 0.09 | 0.77 ± 0.10 |

| Configuration | # Reaction Steps | # Unique Enforced Blocks | Oracle Budget (Wall Time) |
|---|---|---|---|
| TanSim | 2.53 ± 1.3 (M=1665) | 2 ± 0 | 3,000 (5h 3m ± 24m) |
| FMS | 2 ± 1.07 (M=48) | 1.5 ± 0.5 | 3,000 (4h 29m ± 17m) |
| TANGO-FMS | 2.27 ± 1.28 (M=2382) | 1.2 ± 0.4 | 3,000 (5h 28m ± 45m) |

## E.2. Fuzzy Matching Substructure is an Asymmetric Reward Function

The FMS results from the previous section yielded *false positives*: A maximum reward (1.0) was assigned to many generated molecules, yet these molecules did not contain any of the enforced building blocks in its synthesis graphs. The reason for this is due to the asymmetric nature of the designed FMS reward function. We refer to Fig. 7. The FMS reward function computes the maximum substructure overlap and then divides the number of atoms in this overlap by the number of atoms in the enforced building block. Fig. 7 illustrates an edge case where the intermediate node contains the enforced building block as a substructure, but the overall structures do not exactly match. The result was that FMS assigned a perfect reward (1.0). This edge case can be handled by an additional check for exact match, and returning the asymmetric FMS otherwise. This is one possible solution to avoid false positives, yet still reward the model since the overall node and enforced building block structures are similar. **Therefore, for all FMS reward function results, we used this formulation.**

However, we note that false positives only occur in the FMS reward function case, as TANGO-FMS *cannot* yield perfect reward. Since TANGO-FMS is comprised of both FMS and Tanimoto similarity: even if FMS is a false positive, Tanimoto similarity cannot equal 1.0, and thus TANGO-FMS cannot equal 1.0. We hypothesized that this false positive can actually be beneficial, as a perfect reward biases the model towards generating molecules that yield a synthesis graph with an intermediate node *similar* to an enforced building block. This exploitation behavior could be advantageous. In the next section, we investigate the exploration-exploitation trade-off of the generative model when using TANGO-FMS as the reward function. Once we identified optimal hyperparameters, we performed an ablation study in the section after to quantitatively study this asymmetric FMS behavior. We sought to answer whether it is *actually* advantageous to return a "perfect reward (1.0)" for the FMS component in these situations?

## E.3. Can we Circumvent Reward Stagnation?

The results from the first section identified TANGO-FMS as the most stable reward function. However, during RL, we observed that the reward improvement often stagnates.

**Hypotheses:**

1. Further relax the local sampling behavior of Saturn which may help reward stagnation

**Fixed Parameters:**

1. **Oracle Budget** = 5,000

2. **Batch Size** = 32 or 64

**Fuzzy Matching Substructure (*Asymmetric Relationship*)**

**Intermediate Node**               **Enforced Building Block (EBB)**

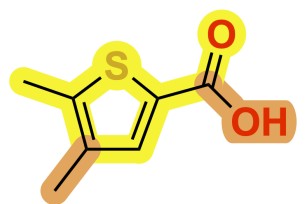   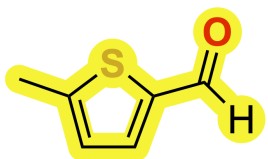

**Number of Overlapping Heavy Atoms:** 8

**Normalized by EBB:** $\frac{8}{8} = \mathbf{1.0}$

**Normalized by Node:** $\frac{8}{10} = \mathbf{0.8}$

*Figure 7.* Fuzzy Matching Substructure (FMS) is asymmetric depending on whether the number of matching atoms is divided by the number of atoms in the enforced building block or the intermediate node.

3. **Augmentation Rounds** = Varied

4. **Enforced Building Blocks** = 100

**Observations:** Table 4 shows the results with the mean and standard deviation across 5 seeds (0-4 inclusive). We make the following observations:

1. Batch32, AR5 is the most successful but imposes a *much longer* wall time. This is due to Saturn's local sampling behavior at low batch sizes and high augmentation rounds.

2. Batch64, AR0 is essentially completely unsuccessful. This affirms that some degree of exploitation is beneficial.

3. Batch64, AR10 is somewhat inconsistent, suggesting *too much* exploitation.

4. Batch64, AR2 and AR5 performs well with the latter notably better, suggesting AR5 may be a good balance between exploration-exploitation.

**E.4. Re-visiting Reward Function Formulation for Ablation Studies**

The results from the previous section identified tentative hyperparameters with agood balance between exploration-exploitation. With this "better" sampling behavior, we wanted to re-visit the reward function formulations as an extensive ablation to affirm that TANGO-FMS is the best formulation.

**Hypotheses:**

1. TANGO-FMS may not be the best reward function formulation now that better exploration-exploitation parameters have been identified. Try all reward function formulations.

2. In the previous section, Batch64, AR5 worked much better than Batch64, AR2, but it *might* be too exploitative when we consider moving to a smaller set of enforced blocks and/or starting-material constraints.

3. More thoroughly study the effect of the sampling behavior by increasing the oracle budget.

*Table 4.* Results for Section 2: Can we Circumvent Reward Stagnation? The mean and standard deviation across 5 seeds (0-4 inclusive) are reported. The number of replicates (out of 5) with at least 1 generated molecule that is synthesizable with an enforced building block is reported with **N**. Batch denotes "Batch Size" and AR denotes "Augmentation Rounds". The number of molecules (pooled across all successful replicates) is partitioned into different docking score thresholds and statistics are reported. # Reaction Steps is also reported for the pooled generated molecules that have an enforced block. The total number of molecules in each pool across the 5 seeds is denoted by **M**. For the docking score intervals, we report the scores and QED values.

| Configuration | Synthesizability | | Docking Score Intervals (QED Annotated) | | |
|---|---|---|---|---|---|
| | Non-solved | Solved (Enforced) | DS < -10 | -10 < DS < -9 | -9 < DS < -8 |
| Batch32, AR5 (N=4) | 1125 ± 110 | 960 ± 871 | -10.34 ± 0.23 (M=11) 0.82 ± 0.08 | -9.41 ± 0.25 (M=220) 0.79 ± 0.11 | -8.41 ± 0.25 (1093) 0.80 ± 0.09 |
| Batch64, AR10 (N=3) | 954 ± 90 | 674 ± 706 | -10.43 ± 0.38 (M=78) 0.68 ± 0.07 | -9.46 ± 0.25 (M=208) 0.71 ± 0.10 | -8.41 ± 0.25 (484) 0.81 ± 0.09 |
| Batch64, AR5 (N=4) | 1029 ± 78 | 857 ± 529 | -10.3 ± 0.25 (M=14) 0.73 ± 0.11 | -9.37 ± 0.21 (M=274) 0.78 ± 0.10 | -8.44 ± 0.25 (M=1210) 0.81 ± 0.09 |
| Batch64, AR2 (N=4) | 1175 ± 89 | 33 ± 47 | -10.7 ± 0 (M=1) 0.75 ± 0 | -9.21 ± 0.10 (M=8) 0.82 ± 0.15 | -8.39 ± 0.25 (M=49) 0.77 ± 0.11 |
| Batch64, AR0 (N=3) | 1921 ± 72 | 0.60 ± 0.49 | None N/A | None N/A | -8.70 ± 0 (M=1) 0.55 ± 0 |

| Configuration | # Reaction Steps | # Unique Enforced Blocks | Oracle Budget (Wall Time) |
|---|---|---|---|
| Batch32, AR5 | 2.76 ± 1.41 (M=4800) | 1 ± 0 | 5,000 (11h 44m ± 1h 38m) |
| Batch64, AR10 | 2.76 ± 1.12 (M=3370) | 2 ± 0.82 | 5,000 (5h 56m ± 25m) |
| Batch64, AR5 | 2.13 ± 1.21 (M=4286) | 1.25 ± 0.43 | 5,000 (4h 21m ± 18m) |
| Batch64, AR2 | 1.60 ± 0.82 (M=1234) | 2 ± 1 | 5,000 (3h 20m ± 7m) |
| Batch64, AR0 | 1 ± 0 (M=3) | 1 ± 0 | 5,000 (2h 52m ± 2m) |

**Fixed Parameters:**

1. **Oracle Budget** = 10,000

2. **Batch Size** = 64

3. **Augmentation Rounds** = 2 or 5

4. **Enforced Building Blocks** = 100

**Observations:** Table 5 shows the results with the mean and standard deviation across 5 seeds (0-4 inclusive). We make the following observations:

1. Surprisingly, Brute-force is sometimes successful but is inconsistent, as expected. Notably many molecules are non-solved (no retrosynthesis route found).

2. FG poorly distinguishes between "goodness" and is essentially unsuccessful, as expected.

3. FMS can distinguish between "goodness" but is not very successful, somewhat unexpectedly.

4. TamSim continues to be successful but is inconsistent, mirroring initial results.

5. TANGO with components of FG are more unsuccessful, in agreement with FG being a poor reward function formulation.

6. TANGO-FMS is most stable, mirroring initial results.

7. TANGO-FMS with the "Asymmetric FMS" implementation (Appendix E.2) performs worse than without. The variance for Solved (Enforced) is higher and much fewer molecules with good docking scores and QED are generated. For this reason, from here on, the original FMS implementation is used, as detailed in Appendix E.2.

8. TANGO-FMS (but with AR5) can outperform TANGO-FMS (AR2) but is notably more inconsistent. This affirms our hypothesis that AR5 might be too exploitative. Importantly, the runs with AR5 also have a much longer wall time, again, due to Saturn's local sampling behavior. Based on these results, AR2 is likely a better balance between exploration-exploitation.

*Table 5.* Results for Section 3: Re-visiting Reward Function Formulation for Ablation Studies. The mean and standard deviation across 5 seeds (0-4 inclusive) are reported. The number of replicates (out of 5) with at least 1 generated molecule that is synthesizable with an enforced building block is reported with **N**. AR denotes "Augmentation Rounds". The number of molecules (pooled across all successful replicates) is partitioned into different docking score thresholds and statistics are reported. # Reaction Steps is also reported for the pooled generated molecules that have an enforced block. The total number of molecules in each pool across the 5 seeds is denoted by **M**. For the docking score intervals, we report the scores and QED values.

| Configuration | Synthesizability | | Docking Score Intervals (QED Annotated) | | |
|---|---|---|---|---|---|
| | Non-solved | Solved (Enforced) | DS < -10 | -10 < DS < -9 | -9 < DS < -8 |
| Brute-force | $5547 \pm 1554$ | $2175 \pm 1954$ | $-10.36 \pm 0.25$ (M=25) | $-9.33 \pm 0.22$ (M=529) | $-8.43 \pm 0.25$ (M=3196) |
| (N=3) | | | $0.46 \pm 0.16$ | $0.60 \pm 0.19$ | $0.69 \pm 0.19$ |
| TanSim | $2275 \pm 143$ | $1863 \pm 1827$ | $-10.36 \pm 0.24$ (M=30) | $-9.36 \pm 0.23$ (M=487) | $-8.43 \pm 0.25$ (M=2389) |
| (N=3) | | | $0.72 \pm 0.09$ | $0.76 \pm 0.10$ | $0.79 \pm 0.09$ |
| FG | $2144 \pm 263$ | $1 \pm 1$ | None | $-9.20 \pm 0$ (M=1) | $-8.50 \pm 0.28$ (M=2) |
| (N=4) | | | N/A | $0.87 \pm 0$ | $0.85 \pm 0.03$ |
| FMS | $1693 \pm 174$ | $114 \pm 205$ | $-10.36 \pm 0.40$ (M=5) | $-9.41 \pm 0.25$ (M=53) | $-8.48 \pm 0.25$ (M=173) |
| (N=5) | | | $0.75 \pm 0.15$ | $0.85 \pm 0.09$ | $0.85 \pm 0.07$ |
| TANGO-FG | $1957 \pm 203$ | $658 \pm 967$ | $-10.20 \pm 0.10$ (M=9) | $-9.27 \pm 0.18$ (M=205) | $-8.48 \pm 0.25$ (M=1280) |
| (N=5) | | | $0.74 \pm 0.07$ | $0.78 \pm 0.10$ | $0.82 \pm 0.09$ |
| TANGO-FMS | $2229 \pm 325$ | $1743 \pm 715$ | $-10.34 \pm 0.25$ (M=218) | $-9.44 \pm 0.25$ (M=1606) | $-8.49 \pm 0.26$ (M=2206) |
| (N=5) | | | $0.77 \pm 0.11$ | $0.83 \pm 0.10$ | $0.82 \pm 0.10$ |
| TANGO-FMS | $2249 \pm 323$ | $1866 \pm 1083$ | $-10.36 \pm 0.27$ (M=59) | $-9.39 \pm 0.25$ (M=513) | $-8.40 \pm 0.25$ (M=2049) |
| Asymmetric-FMS (N=5) | | | $0.71 \pm 0.10$ | $0.79 \pm 0.10$ | $0.80 \pm 0.09$ |
| TANGO-FMS (AR5) | $2157 \pm 182$ | $2521 \pm 2060$ | $-10.29 \pm 0.18$ (M=11) | $-9.33 \pm 0.22$ (M=382) | $-8.40 \pm 0.24$ (M=2881) |
| (N=4) | | | $0.71 \pm 0.12$ | $0.80 \pm 0.11$ | $0.83 \pm 0.09$ |
| TANGO-All | $2049 \pm 93$ | $147 \pm 245$ | $-10.43 \pm 0.27$ (M=31) | $-9.41 \pm 0.26$ (M=227) | $-8.53 \pm 0.25$ (M=283) |
| (N=3) | | | $0.74 \pm 0.08$ | $0.82 \pm 0.09$ | $0.84 \pm 0.09$ |

| Configuration | # Reaction Steps | # Unique Enforced Blocks | Oracle Budget (Wall Time) |
|---|---|---|---|
| Brute-force | $3 \pm 1.54$ (M=10876) | $1.33 \pm 0.47$ | 10,000 (7h 29m $\pm$ 2h 11m) |
| TanSim | $2.18 \pm 1.16$ (M=9319) | $1.33 \pm 0.47$ | 10,000 (9h 11m $\pm$ 48m) |
| FG | $3.33 \pm 1.80$ (M=9) | $1.75 \pm 0.83$ | 10,000 (7h 23m $\pm$ 16m) |
| FMS | $1.78 \pm 1.04$ (M=570) | $1.8 \pm 0.75$ | 10,000 (7h 50m $\pm$ 26) |
| TANGO-FG | $2.21 \pm 1.15$ (M=3237) | $1.8 \pm 0.75$ | 10,000 (8h 29m $\pm$ 25m) |
| TANGO-FMS | $2.35 \pm 1.24$ (M=8719) | $2.2 \pm 0.75$ | 10,000 (8h 12m $\pm$ 15m) |
| TANGO-FMS (Asymmetric-FMS) | $2.24 \pm 1.19$ (M=9334) | $2.2 \pm 0.4$ | 10,000 (8h 42m $\pm$ 26m) |
| TANGO-FMS (AR5) | $2.58 \pm 1.17$ (M=12608) | $1.5 \pm 0.5$ | 10,000 (12h 36m $\pm$ 52m) |
| TANGO-All | $2.74 \pm 1.18$ (M=714) | $2 \pm 0.82$ | 10,000 (8h 11m $\pm$ 12m) |

## E.5. In TANGO-FMS, is either FMS or Tanimoto Similarity more Important?

The results from the previous section identified hyperparameters with good balance between exploration-exploitation. Thus far, all TANGO formulations weight each component equally. The next question we asked was whether certain components were more important?

**Hypotheses:**

1. FMS should be more informative than Tanimoto similarity to inform chemical *reactivity*. Test the effect of components weighting.

**Fixed Parameters:**

1. **Oracle Budget** = 10,000

2. **Batch Size** = 64

3. **Augmentation Rounds** = 2

4. **Enforced Building Blocks** = 100

**Observations:** Table 6 shows the results with the mean and standard deviation across 5 seeds (0-4 inclusive). "High" indicates 0.75 weighting while the other component is 0.25. TANGO-FMS has equal weighting (0.5 FMS, 0.5 Tanimoto). We make the following observations:

1. TANGO-FMS with equal weighting performs the best in the context of MPO as docking scores are better.

2. TANGO-FMS-High-TanSim generates more solved molecules but docking scores are worse. These suggests suggest that MPO is better with TANGO-FMS (equal weighting) and is the reward function we use from here on.

*Table 6.* Results for Section 4: In TANGO-FMS, is either FMS or Tanimoto Similarity more Important? The mean and standard deviation across 5 seeds (0-4 inclusive) are reported. The number of replicates (out of 5) with at least 1 generated molecule that is synthesizable with an enforced building block is reported with **N**. The number of molecules (pooled across all successful replicates) is partitioned into different docking score thresholds and statistics are reported. # Reaction Steps is also reported for the pooled generated molecules that have an enforced block. The total number of molecules in each pool is denoted by **M**. For the docking score intervals, we report the scores and QED values.

| Configuration | Synthesizability | | Docking Score Intervals (QED Annotated) | | |
|---|---|---|---|---|---|
| | Non-solved | Solved (Enforced) | DS < -10 | -10 < DS < -9 | -9 < DS < -8 |
| TANGO | $2229 \pm 325$ | $1743 \pm 715$ | $-10.34 \pm 0.25$ (M=218) | $-9.44 \pm 0.25$ (M=1606) | $-8.49 \pm 0.26$ (M=2206) |
| (0.50 FMS, 0.50 TanSim) (N=5) | | | $0.77 \pm 0.11$ | $0.83 \pm 0.10$ | $0.82 \pm 0.10$ |
| TANGO | $1962 \pm 166$ | $1725 \pm 747$ | $-10.30 \pm 0.17$ (M=23) | $-9.32 \pm 0.22$ (M=498) | $-8.44 \pm 0.25$ (M=2554) |
| (0.75 FMS, 0.25 TanSim) (N=5) | | | $0.78 \pm 0.06$ | $0.83 \pm 0.07$ | $0.85 \pm 0.07$ |
| TANGO | $2464 \pm 437$ | $2737 \pm 1038$ | $-10.31 \pm 0.24$ (M=84) | $-9.39 \pm 0.25$ (M=837) | $8.43 \pm 0.25$ (M=3468) |
| (0.25 FMS, 0.75 TanSim) (N=5) | | | $0.75 \pm 0.10$ | $0.78 \pm 0.10$ | $0.80 \pm 0.10$ |

| Configuration | # Reaction Steps | # Unique Enforced Blocks | Oracle Budget (Wall Time) |
|---|---|---|---|
| TANGO (0.5 FMS, 0.5 TanSim) | $2.35 \pm 1.24$ (M=8719) | $2.2 \pm 0.75$ | 10,000 (8h 12m $\pm$ 15m) |
| TANGO (0.75 FMS, 0.25 TanSim) | $2.39 \pm 1.24$ (M=8625) | $1.6 \pm 0.49$ | 10,000 (8h 36m $\pm$ 16m) |
| TANGO (0.25 FMS, 0.75 TanSim) | $2.30 \pm 1.30$ (M=13688) | $1.6 \pm 0.49$ | 10,000 (8h 51m $\pm$ 26m) |

## E.6. Investigating Robustness

With optimal hyperparameters identified, we expand to robustness studies and run every experiment across 10 seeds (0-9 inclusive) and investigate enforcing a smaller set of building blocks. We also probe whether the starting-material constraint is also learnable within the oracle budget. Finally, we also perform a set of experiments *without* the QED objective.

**Fixed Parameters:**

1. **Oracle Budget** = 10,000

2. **Batch Size** = 64

3. **Augmentation Rounds** = 2

4. **Reward Function** = TANGO-FMS (equal weighting)

5. **Enforced Building Blocks** = 100

**Observations:** Table 7 shows the results with the mean and standard deviation across 10 seeds (0-9 inclusive). We make the following observations:

1. All constraints are learnable.

2. When not enforcing QED, the model generates many more molecules with "good" docking scores, and expectedly, at the expense of QED. This affirms that the MPO is tunable, allowing tailored design of molecules that are also constrained by synthesis.

3. As expected, when not enforcing QED, the average reaction steps is longer, since QED constrains molecular weight.

*Table 7.* Results for Section 5: Investigating Robustness. "SM" denotes starting-material constrained. The mean and standard deviation across 10 seeds (0-9 inclusive) are reported. The number of replicates (out of 10) with at least 1 generated molecule that is synthesizable with an enforced building block is reported with **N**. TThe number of molecules (pooled across all successful replicates) is partitioned into different docking score thresholds and statistics are reported. # Reaction Steps is also reported for the pooled generated molecules that have an enforced block. The total number of molecules in each pool across the 10 seeds is denoted by **M**. For the docking score intervals, we report the scores and QED values.

| Configuration | Synthesizability | | Docking Score Intervals (QED Annotated) | | |
|---|---|---|---|---|---|
| | Non-solved | Solved (Enforced) | DS $<$ -10 | -10 $<$ DS $<$ -9 | -9 $<$ DS $<$ -8 |
| 100 Blocks | 2288 $\pm$ 305 | 2111 $\pm$ 1169 | -10.36 $\pm$ 0.28 (M=487) | -9.42 $\pm$ 0.24 (M=3096) | -8.47 $\pm$ 0.26 (M=5904) |
| (N=10) | | | 0.79 $\pm$ 0.09 | 0.82 $\pm$ 0.09 | 0.81 $\pm$ 0.09 |
| 100 Blocks (no QED) | 1848 $\pm$ 158 | 3723 $\pm$ 681 | -10.74 $\pm$ 0.55 (M=8649) | -9.48 $\pm$ 0.26 (M=10633) | -8.53 $\pm$ 0.25 (M=9771) |
| (N=10) | | | 0.23 $\pm$ 0.06 | 0.27 $\pm$ 0.11 | 0.32 $\pm$ 0.15 |
| 100 Blocks (SM) | 1879 $\pm$ 186 | 1524 $\pm$ 502 | -10.41 $\pm$ 0.31 (M=120) | -9.41 $\pm$ 0.24 (M=985) | -8.43 $\pm$ 0.25 (M=3156) |
| (N=10) | | | 0.78 $\pm$ 0.09 | 0.81 $\pm$ 0.09 | 0.81 $\pm$ 0.09 |
| 100 Blocks (SM, no QED) | 1734 $\pm$ 172 | 1189 $\pm$ 963 | -10.50 $\pm$ 0.40 (M=685) | -9.43 $\pm$ 0.25 (M=2357) | -8.49 $\pm$ 0.25 (M=4121) |
| (N=10) | | | 0.31 $\pm$ 0.15 | 0.38 $\pm$ 0.17 | 0.45 $\pm$ 0.18 |
| 10 Blocks | 2425 $\pm$ 288 | 984 $\pm$ 1181 | -10.38 $\pm$ 0.30 (M=659) | -9.46 $\pm$ 0.25 (M=3981) | -8.57 $\pm$ 0.25 (M=2419) |
| (N=6) | | | 0.79 $\pm$ 0.10 | 0.83 $\pm$ 0.09 | 0.83 $\pm$ 0.10 |
| 10 Blocks (no QED) | 1967 $\pm$ 211 | 2640 $\pm$ 1066 | -10.51 $\pm$ 0.39 (M=3453) | -9.47 $\pm$ 0.25 (M=8402) | -8.54 $\pm$ 0.25 (M=8332) |
| (N=9) | | | 0.35 $\pm$ 0.16 | 0.39 $\pm$ 0.16 | 0.41 $\pm$ 0.16 |
| 10 Blocks (SM) | 2228 $\pm$ 182 | 1004 $\pm$ 925 | -10.37 $\pm$ 0.27 (794) | -9.46 $\pm$ 0.24 (M=3881) | -8.54 $\pm$ 0.25 (M=2790) |
| (N=9) | | | 0.80 $\pm$ 0.09 | 0.83 $\pm$ 0.09 | 0.84 $\pm$ 0.10 |
| 10 Blocks (SM, no QED) | 1753 $\pm$ 147 | 1563 $\pm$ 1111 | -10.57 $\pm$ 0.45 (M=2439) | -9.47 $\pm$ 0.25 (M=5120) | -8.54 $\pm$ 0.25 (M=4649) |
| (N=8) | | | 0.35 $\pm$ 0.15 | 0.43 $\pm$ 0.17 | 0.44 $\pm$ 0.17 |

| Configuration | # Reaction Steps | # Unique Enforced Blocks | Oracle Budget (Wall Time) |
|---|---|---|---|
| 100 Blocks | 2.37 $\pm$ 1.27 (M=21115) | 2 $\pm$ 0.63 | 10,000 (8h 31m $\pm$ 40m) |
| 100 Blocks (no QED) | 3.24 $\pm$ 1.20 (M=37231) | 1.8 $\pm$ 0.75 | 10,000 (7h 27m $\pm$ 13m) |
| 100 Blocks (SM) | 1.49 $\pm$ 0.91 (M=15247) | 1.9 $\pm$ 0.7 | 10,000 (8h 33m $\pm$ 30m) |
| 100 Blocks (SM, no QED) | 2.34 $\pm$ 1.18 (M=11890) | 1.7 $\pm$ 0.64 | 10,000 (8h 3m $\pm$ 33m) |
| 10 Blocks | 2.70 $\pm$ 1.20 (M=9845) | 1 $\pm$ 0 | 10,000 (8h 29m $\pm$ 30m) |
| 10 Blocks (no QED) | 3.18 $\pm$ 1.25 (M=26403) | 1.22 $\pm$ 0.42 | 10,000 (7h 51m $\pm$ 33m) |
| 10 Blocks (SM) | 2.59 $\pm$ 1.04 (M=10040) | 1 $\pm$ 0 | 10,000 (8h 39m $\pm$ 24m) |
| 10 Blocks (SM, no QED) | 2.65 $\pm$ 0.88 (M=15632) | 1 $\pm$ 0 | 10,000 (8h 9m $\pm$ 27m) |

### E.7. Lucky Building Blocks?

From the previous set of experiments, we noticed that the generative model was always incorporating the same 3 enforced building blocks. One in particular was especially common, such that most runs using the set of *10* enforced blocks, use it. We questioned whether TANGO's success was due to luck in having "suitable" building blocks. Therefore, we perform further ablation experiments that purge these 3 building blocks. Similar to the previous set of experiments, we run every configuration here across 10 seeds (0-9 inclusive).

**Fixed Parameters:**

1. **Oracle Budget** = 10,000

2. **Batch Size** = 64

3. **Augmentation Rounds** = 2

4. **Reward Function** = TANGO-FMS (equal weighting)

5. *Purged* **Enforced Building Blocks** = 97 (Purged the 3 common enforced building blocks from the set of 100)

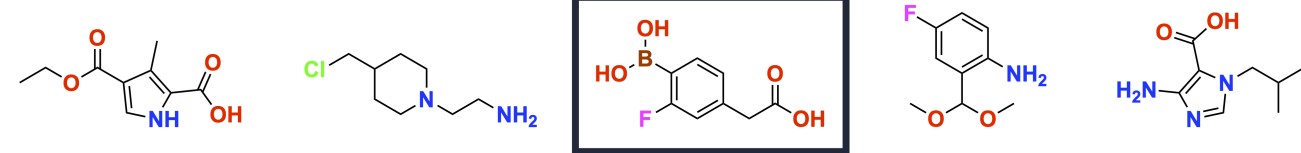

*Figure 8.* 5 Enforced Building Blocks Set. The circled block is the Suzuki coupling reagent used in all the successful runs without QED (N=2/10 seeds).

**Observations:** Table 8 shows the results with the mean and standard deviation across 10 seeds (0-9 inclusive). We make the following observations:

1. Other building blocks can be enforced.

2. The runs become less consistent (less successful seeds out of 10). Runs without QED are consistently succcessful, suggesting that the *commonly* enforced blocks were chosen due to being able to jointly satisfy QED and docking.

*Table 8.* Results for Section 6: Lucky Building Blocks? "SM" denotes starting-material constrained. The mean and standard deviation across 10 seeds (0-9 inclusive) are reported. The number of replicates (out of 10) with at least 1 generated molecule that is synthesizable with an enforced building block is reported with **N**. The number of molecules (pooled across all successful replicates) is partitioned into different docking score thresholds and statistics are reported. # Reaction Steps is also reported for the pooled generated molecules that have an enforced block. The total number of molecules in each pool across the 10 seeds is denoted by **M**. For the docking score intervals, we report the scores and QED values.

| Configuration | Synthesizability | | Docking Score Intervals (QED Annotated) | | |
|---|---|---|---|---|---|
| | Non-solved | Solved (Enforced) | DS < -10 | -10 < DS < -9 | -9 < DS < -8 |
| 100 Blocks Purged | 2322 ± 233 | 77 ± 229 | -10.56 ± 0.37 (M=17) | -9.38 ± 0.24 (M=117) | -8.48 ± 0.25 (M=376) |
| (N=4) | | | 0.85 ± 0.03 | 0.87 ± 0.05 | 0.89 ± 0.04 |
| 100 Blocks Purged (no QED) | 1794 ± 193 | 1553 ± 1211 | -10.65 ± 0.49 (M=2649) | -9.47 ± 0.26 (M=4345) | -8.52 ± 0.25 (M=4648) |
| (N=9) | | | 0.25 ± 0.11 | 0.30 ± 0.15 | 0.36 ± 0.17 |
| 100 Blocks Purged (SM) | 2179 ± 298 | 166 ± 333 | -10.30 ± 0.17 (M=6) | -9.39 ± 0.25 (M=128) | -8.44 ± 0.24 (M=636) |
| (N=5) | | | 0.83 ± 0.08 | 0.83 ± 0.09 | 0.87 ± 0.07 |
| 100 Blocks Purged (SM, no QED) | 1688 ± 239 | 1456 ± 1112 | -10.49 ± 0.38 (M=1032) | -9.43 ± 0.25 (M=3871) | -8.52 ± 0.25 (M=5624) |
| (N=8) | | | 0.34 ± 0.11 | 0.36 ± 0.13 | 0.37 ± 0.15 |

| Configuration | # Reaction Steps | # Unique Enforced Blocks | Oracle Budget (Wall Time) |
|---|---|---|---|
| 100 Blocks Purged | 5.97 ± 1.17 (M=769) | 1.25 ± 0.43 | 10,000 (8h 54m ± 20m) |
| 100 Blocks Purged (no QED) | 3.35 ± 1.19 (M=15525) | 1.44 ± 0.50 | 10,000 (7h 15m ± 12m) |
| 100 Blocks Purged (SM) | 4.12 ± 2.29 (M=1660) | 1.2 ± 0.4 | 10,000 (8h 41m ± 28m) |
| 100 Blocks Purged (SM, no QED) | 3.30 ± 1.28 (M=14562) | 1.62 ± 0.70 | 10,000 (7h 40m ± 25m) |

### E.8. 5 Enforced Blocks

We next push our framework further by curating 5 building blocks (Fig. 8) that are dissimilar and/or can be involved in *different* reaction chemistries. Our objective was to investigate whether the model can learn to incorporate such a small set of blocks and whether other chemical reactions can be enforced.

**Fixed Parameters:**

1. **Oracle Budget** = 10,000 or 15,000

2. **Batch Size** = 64

3. **Augmentation Rounds** = 2

4. **Reward Function** = TANGO-FMS (equal weighting)

5. **Enforced Building Blocks** = 5 (dissimilar to the ones used thus far)

**Observations:** Table 9 shows the results with the mean and standard deviation across 10 seeds (0-9 inclusive). We make the following observations:

1. The task is challenging under the 10,000 oracle budget when QED is also optimized for.

2. Without optimizing for QED and increasing the oracle budget to 15,000 results in some successes (2/10 seeds).

3. The two successful replicates both enforced only the Suzuki block (Boron containing) which is circled in Fig. 8.

4. The results here show that learning to enforce such a small set of building blocks is possible. In practice, one could further increase the oracle budget which we did not explore due to time limits on the cluster we used. The two successful replicates (with a 15,000 oracle budget) took about 12.5 hours which we believe is still very reasonable.

*Table 9.* Results for Section 7: 5 Enforced Blocks. The mean and standard deviation across 10 seeds (0-9 inclusive) are reported. The number of replicates (out of 10) with at least 1 generated molecule that is synthesizable with an enforced building block is reported with **N**. The number of molecules (pooled across all successful replicates) is partitioned into different docking score thresholds and statistics are reported. # Reaction Steps is also reported for the pooled generated molecules that have an enforced block. The total number of molecules in each pool is denoted by **M**. For the docking score intervals, we report the scores and QED values.

| Configuration | Synthesizability | | Docking Score Intervals (QED Annotated) | | |
|---|---|---|---|---|---|
| | Non-solved | Solved (Enforced) | DS < -10 | -10 < DS < -9 | -9 < DS < -8 |
| 5 Blocks | $2639 \pm 186$ | $0 \pm 0$ | N/A | N/A | N/A |
| (N=0) | | | N/A | N/A | N/A |
| 5 Blocks (no QED, 15k Budget) | $3333 \pm 437$ | $972 \pm 2112$ | $-11.73 \pm 0.93$ (M=7044) | $-9.51 \pm 0.26$ (M=1419) | $-8.59 \pm 0.24$ (M=670) |
| (N=2) | | | $0.29 \pm 0.07$ | $0.38 \pm 0.13$ | $0.39 \pm 0.16$ |

| Configuration | # Reaction Steps | # Unique Enforced Blocks | Oracle Budget (Wall Time) |
|---|---|---|---|
| 5 Blocks | N/A | N/A | 10,000 (9h 24m $\pm$ 25m) |
| 5 Blocks (no QED, 15k Budget) | $3.79 \pm 0.83$ (M=9723) | $1 \pm 0$ | 15,000 (12h 33m $\pm$ 34m) |

### E.9. Divergent Synthesis

Often, divergent synthesis (Li et al., 2018) is desirable, whereby intermediates (usually non-commercially available) are enforced in the synthesis path. This can be used for late-stage functionalization (Castellino et al., 2023) which is particularly relevant in drug discovery to explore SAR. In this section, we select intermediate non-commercial blocks from solved paths. We note that this is artificial in the sense that these selected intermediates were taken from solved routes, and are likely "favorable". However, we were interested in whether a model can learn *from scratch* to enforce relatively large building blocks. For this reason, we curated 10 selected intermediates and investigate the ability of TANGO to learn divergent synthesis constraints.

**Fixed Parameters:**

1. **Oracle Budget** = 10,000 or 15,000

2. **Batch Size** = 64

3. **Augmentation Rounds** = 2

4. **Reward Function** = TANGO-FMS (equal weighting)

5. *Divergent* **Enforced Building Blocks** = 10 (Curated from successful runs)

**Observations:** Table 10 shows the results with the mean and standard deviation across 10 seeds (0-9 inclusive). We make the following observations:

1. Divergent blocks can be enforced but the runs are less consistently successful than with the original sets of enforced building blocks, under a 10,000 oracle budget.

2. The runs do not necessarily take longer which means that in practical applications, one could increase the oracle budget. We believe that the wall times of all our experiments (7-9 hours) are reasonable and that much longer is tolerable in real-world applications (<= 24h and even > 24h if the model can truly solve the MPO task).

3. Increasing the oracle budget to 15,000 results in more successful seeds. Therefore, simply using more compute (within reason) is a straightforward solution.

*Table 10.* Results for Section 8: Divergent Synthesis. The mean and standard deviation across 10 seeds (0-9 inclusive) are reported. The number of replicates (out of 10) with at least 1 generated molecule that is synthesizable with an enforced building block is reported with **N**. The number of molecules (pooled across all successful replicates) is partitioned into different docking score thresholds and statistics are reported. # Reaction Steps is also reported for the pooled generated molecules that have an enforced block. The total number of molecules in each pool across the 10 seeds is denoted by **M**. For the docking score intervals, we report the scores and QED values.

| Configuration | Synthesizability | | Docking Score Intervals (QED Annotated) | | |
| --- | --- | --- | --- | --- | --- |
| | Non-solved | Solved (Enforced) | DS < -10 | -10 < DS < -9 | -9 < DS < -8 |
| Divergent Blocks (N=4) | 2166 ± 202 | 651 ± 1238 | -10.36 ± 0.26 (M=187) 0.84 ± 0.10 | -9.41 ± 0.24 (M=1311) 0.86 ± 0.07 | -8.48 ± 0.25 (M=2694) 0.86 ± 0.07 |
| Divergent Blocks (15k Budget) (N=5) | 3720 ± 631 | 1519 ± 2321 | -10.36 ± 0.25 (M=538) 0.82 ± 0.08 | -9.44 ± 0.25 (M=3191) 0.85 ± 0.08 | -8.47 ± 0.25 (M=6324) 0.87 ± 0.08 |
| Divergent Blocks (no QED) (N=3) | 1937 ± 210 | 540 ± 1259 | -10.61 ± 0.47 (M=1099) 0.29 ± 0.11 | -9.48 ± 0.25 (M=1894) 0.41 ± 0.18 | -8.56 ± 0.26 (M=1518) 0.52 ± 0.22 |
| Divergent Blocks (no QED, 15k Budget) (N=4) | 2866 ± 523 | 839 ± 1972 | -10.57 ± 0.42 (M=1861) 0.32 ± 0.13 | -9.48 ± 0.26 (M=3058) 0.40 ± 0.18 | -8.55 ± 0.22 (M=2154) 0.48 ± 0.21 |

| Configuration | # Reaction Steps | # Unique Enforced Blocks | Oracle Budget (Wall Time) |
| --- | --- | --- | --- |
| Divergent Blocks | 3.68 ± 1.08 (M=6512) | 1.75 ± 0.83 | 10,000 (8h 52m ± 42m) |
| Divergent Blocks (15k Budget) | 3.61 ± 1.11 (M=15190) | 1.80 ± 0.17 | 15,000 (15h 54m ± 1h 30m) |
| Divergent Blocks (no QED) | 4.14 ± 1.36 (M=5397) | 1 ± 0 | 10,000 (7h 39m ± 23m) |
| Divergent Blocks (no QED, 15k Budget) | 4.30 ± 1.36 (M=8393) | 1.75 ± 0.83 | 15,000 (12h 41m ± 25m) |

# F. Retrosynthesis Model: Synthesis Routes

In this section, we show examples of synthetic routes from the MEGAN (Sacha et al., 2021) retrosynthesis model with enforced building blocks. The synthesis graph images were taken as is from Syntheseus' (Maziarz et al., 2023) output. Each figure in this section is from an experiment with a different enforced block set (100, 100 with "lucky" blocks purged, 10, 5, and divergent). Moreover, all routes will be shown for molecules with docking score < -10.5 since these are the most optimal. In addition, for the *5 Enforced Blocks*, the routes shown were from the runs without QED. This is because these were the only seeds that were successful under the oracle budget. **The enforced block is boxed.** We also try to show some diversity in the route lengths to highlight that path length was not explicitly optimized for. QED implicitly encourages shorter paths due to constraining the molecular weight, but even so, longer synthetic routes can still be observed (for example in Fig. 10). Future work could also reward shorter paths.

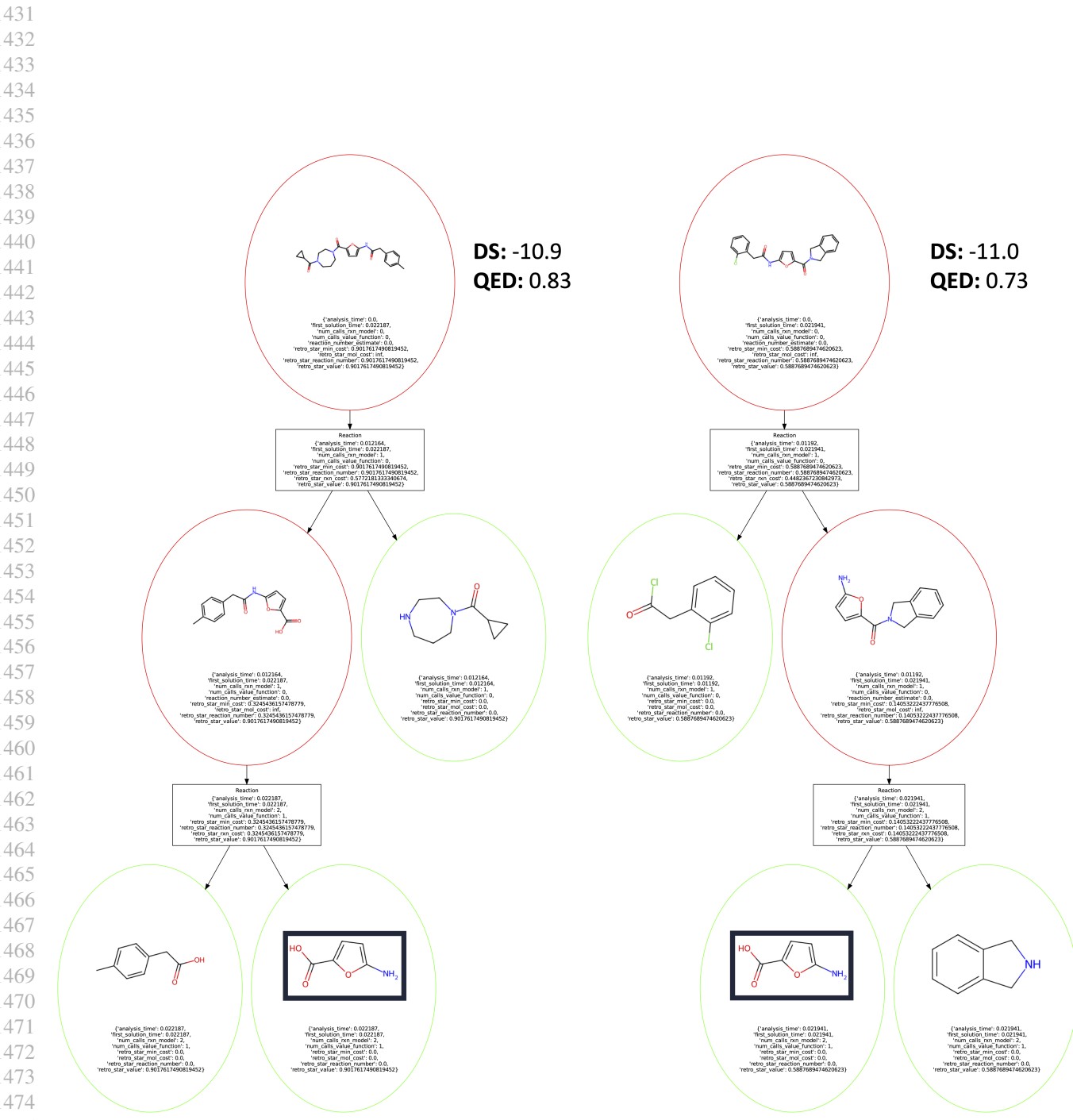

*Figure 9.* 100 Enforced Blocks example routes. The enforced block is boxed.

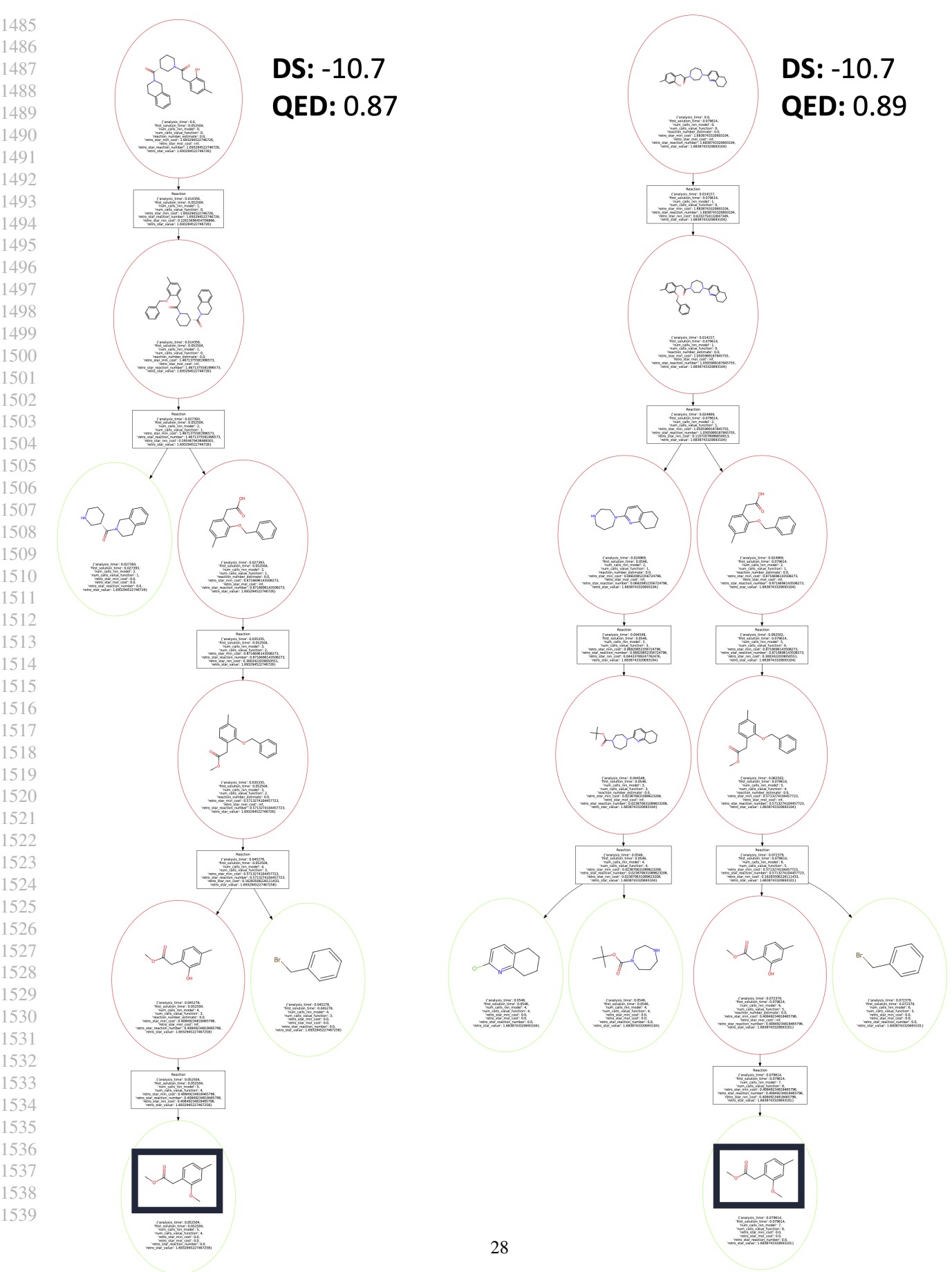

DS: -10.7
QED: 0.87

DS: -10.7
QED: 0.89

*Figure 10.* 100 Enforced Blocks Purged example routes. The enforced block is boxed.

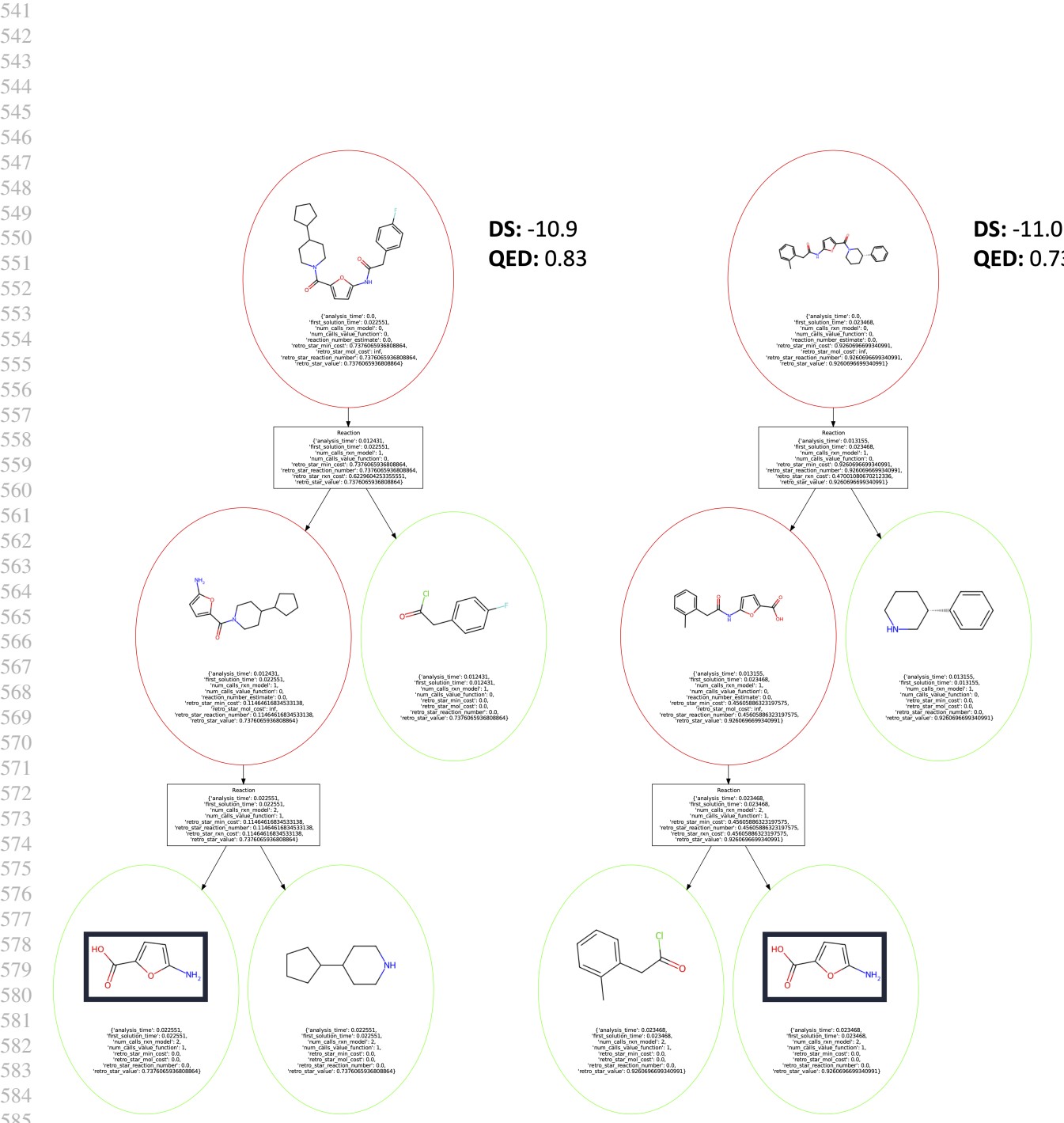

*Figure 11.* 10 Enforced Blocks example routes. The enforced block is boxed.

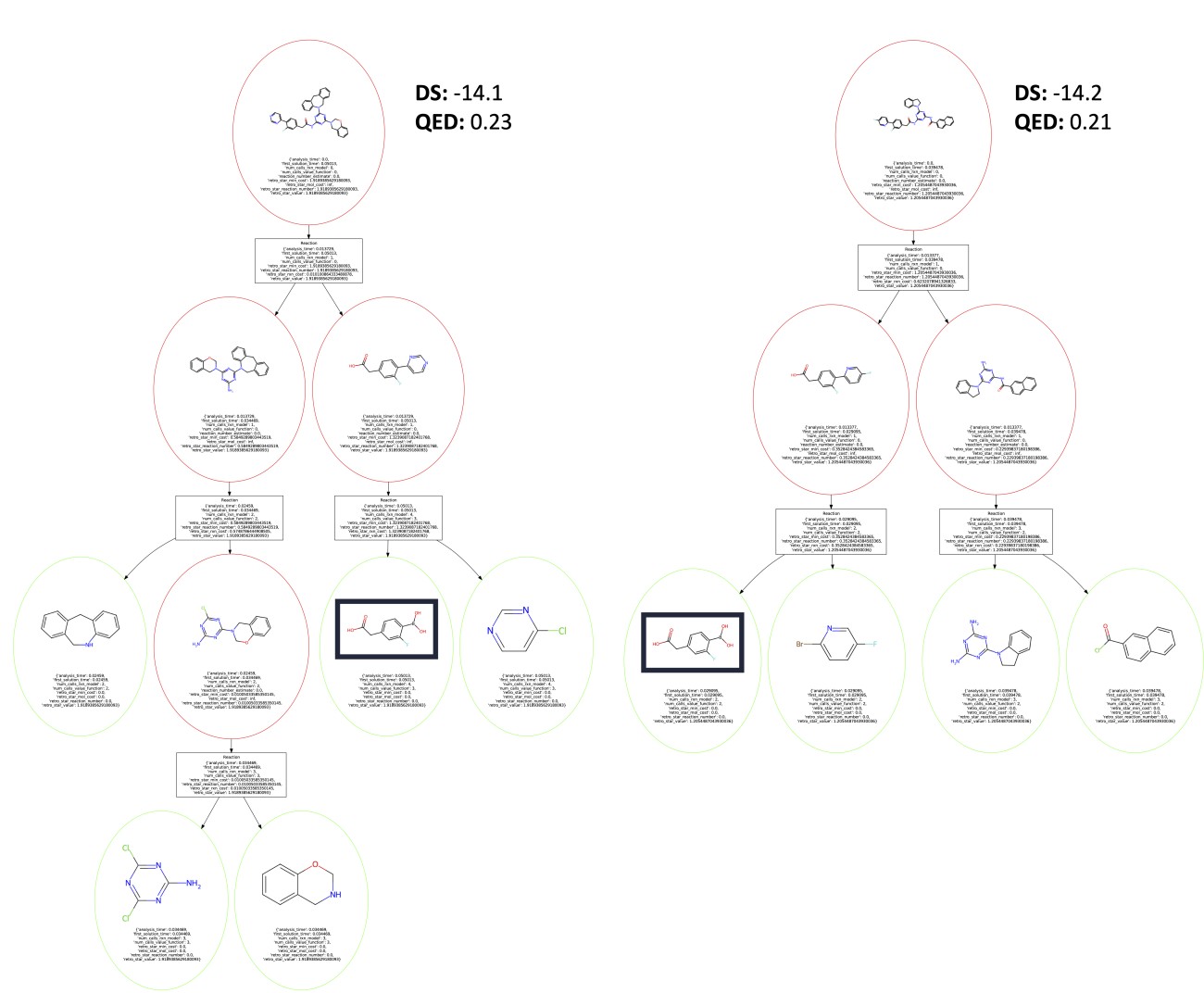

*Figure 12.* 5 Enforced Blocks example routes. Note that QED was not enforced here as the QED experiments did not successful generate any enforced blocks under the oracle budget. The enforced block is boxed.

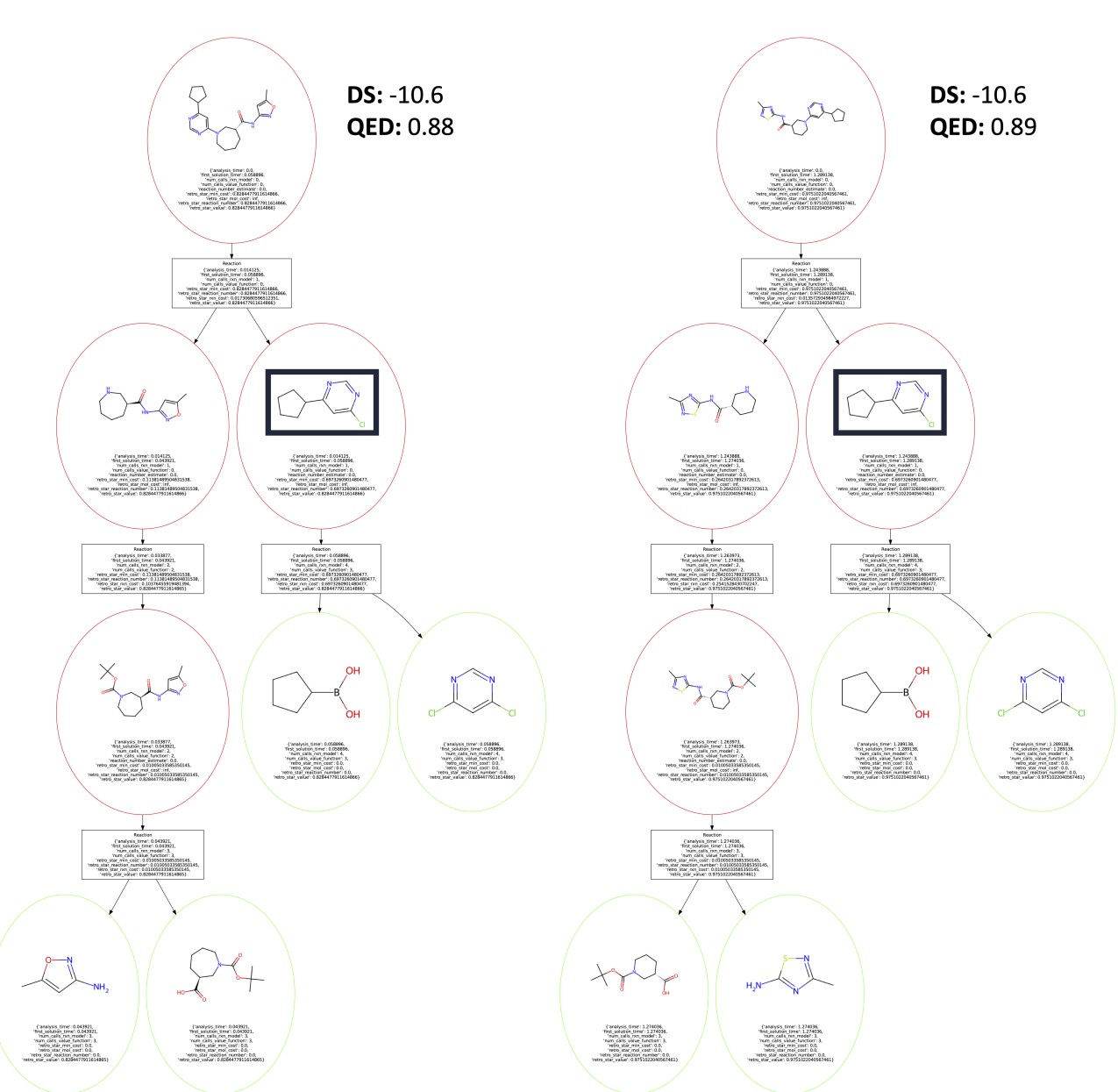

*Figure 13.* Divergent Enforced Blocks example routes. The enforced block is boxed.

