# OpenReview forum: "It Takes Two to Tango: Directly Optimizing for Constrained Synthesizability in Generative Molecular Design"
_ICML.cc/2025/Conference — Submitted to ICML 2025_

### Official Review · Reviewer_337C · 2025-02-15

**Overall Recommendation:** 1

**Summary:**

This work aims to generate synthesizable molecules that meet multi-parameter optimization (MPO) objectives while simultaneously adhering to a predefined set of building blocks. They design reward functions based on chemistry principles and introduce the TANGO reward function to generate synthesizable molecules with enforced building blocks using reinforcement learning (RL).

**Claims And Evidence:**

1. On the other hand, retrosynthesis planning (Liu et al., 2017; Segler & Waller, 2017; Coley et al., 2017; Segler et al., 2018) proposes viable synthetic routes to a target molecule, and these models are often used as stand-alone tools to assess synthesizability

Retrosynthetic planning consists of two key components: a one-step retrosynthesis model and a search algorithm. I believe this claim should be revised for greater accuracy and clarity to avoid potential misunderstandings.

2. Case 1: Starting-material Constrained Synthesis. A synthesis graph is considered starting-material constrained if at least one leaf node, m ∈ G(M, R), satisfies both of the following conditions: (1) m = b ∈ Benf , and (2) depth(m) = max depth:

Case 1 is trivial. In retrosynthetic planning, all leaf nodes of a synthetic route must be starting materials; otherwise, the search is unsuccessful. Therefore, I do not find the constrained synthesis proposed in this work to be novel or innovative.


3. In the context of generative molecular design, previous work has shown that retrosynthesis models can be treated as an oracle and directly optimized for (Guo & Schwaller, 2024c).

In this context, "oracle" refers to an idealized or authoritative source of truth that provides highly reliable or correct answers. Treating retrosynthesis models as an oracle implies assuming they have near-perfect knowledge and decision-making capabilities in predicting retrosynthetic routes. However, in reality, these models have limitations and uncertainties.

In fact, the top-1 accuracy of existing retrosynthesis models on USPTO-50K is typically only 50%-60%, meaning that a significant portion of predictions are incorrect. Given this, it is unreasonable to treat retrosynthesis models as oracles.

**Essential References Not Discussed:**

N/A

**Experimental Designs Or Analyses:**

I check the experimental design and think it's correct.

**Methods And Evaluation Criteria:**

This paper focuses on the synthesizable drug design and uses the solvable rate for evaluation. It makes sense for the problem.

**Other Comments Or Suggestions:**

Use post-training instead of RL from scratch.

**Other Strengths And Weaknesses:**

Weaknesses:

1. Coming from the machine learning community, I would like to clarify why this paper adopts $B_{ref}$ instead of using the entire building block space. The authors employ reinforcement learning (RL) for molecular generation, and if the building block space is too large, the reward signal during training becomes extremely sparse. This leads to low sampling efficiency, making it difficult to identify viable routes for optimization. By constraining the building block space to a smaller subset, the reward signal becomes denser, stabilizing the RL training process.

However, restricting the building block space in this way significantly compromises the model's generalization ability. I do not believe this approach enables the proposed method to be truly applicable in practice.

Furthermore, training RL from scratch essentially resembles a search strategy. In large language models (LLMs), RL is typically applied on top of a pretrained model rather than from scratch. A key factor in stabilizing RL training is having a strong initial starting point. Given this, I find the approach adopted in this paper unpromising.


2. The TANimoto Group Overlap (TANGO) reward function proposed in this paper closely resembles the Process Reward Model (PRM) in LLMs, which assigns rewards based on the prediction process. However, recent approaches, such as the outcome reward model used in DeepSeek R1 [1], do not reward the process itself, as LLMs can easily exploit PRMs. The paper does not discuss the potential limitations of its proposed reward function.

3. It is incorrect to treat the retrosynthesis model as an oracle. Recent work [2] has shown that these models can generate phantom predictions, highlighting their inherent uncertainties.


[1] DeepSeek-R1: Incentivizing Reasoning Capability in LLMs via Reinforcement Learning.

[2] Retro-fallback: Retrosynthetic Planning in an Uncertain World.


---

update: I can't accept this paper currently.

**Questions For Authors:**

N/A

**Relation To Broader Scientific Literature:**

The key contribution is related to our domain (synthesizable drug degisn).

**Theoretical Claims:**

No theoretical analysis.

---

> ### Author Rebuttal · Authors · 2025-03-29
>
> We thank the reviewer for their constructive feedback. We respectively suggest that our work has been misinterpreted and we want to clarify our framework.
>
> ## **1. Clarify retrosynthesis uses single-step model + search algorithm**
>
> We will clarify this in the updated manuscript.
>
> ## **2. Starting-material constraint is trivial, not novel, nor innovative**
>
> We want to clarify what starting-material constraint means. The reviewer points out that in retrosynthesis, all leaf nodes are starting materials. However, all leaf nodes only need to be **commercially available** for a search to be successful. We denote commercially available nodes as $B_{stock}$ (this work uses 17.7 million blocks derived from ZINC). Starting-material constraint means that at least 1 leaf node **at the max depth (first step of synthesis)** must be from $B_{enf}$ (in this work, $B_{enf}$ $\in$ [5, 10, 100] and $B_{enf}$ $\subset$ $B_{stock}$). **This is non-trivial because it heavily constrains the search**. In our notation, we use $m = b \in B_{enf}$ and $depth(m)$ = max depth, as the reviewer also writes.
>
> ## **3. Retrosynthesis models are not oracles because they are not authoritative and top-1 accuracy is low**
>
> We believe the fact that “oracles” are imperfect does not preclude its use as a term. In one of the most popular benchmarks in drug discovery, the Practical Molecular Optimization benchmark [1], the term oracle is introduced as any computational predictor. In that work, surrogate models are considered oracles, even though they are trained with limited data so they are not expected to have global coverage. For retrosynthesis models, the reviewer rightly points out top-1 accuracy is imperfect. However, top-1 accuracy itself is an imperfect measure of oracle effectiveness as **there are many ways to make the same molecule.** However, how would one assess the usefulness of such alternative pathways? We believe it is most convincing to reference real life medicinal chemistry case studies, for example presented in this paper [2], where retrosynthesis models positively impacted 7 commercial drug discovery synthesis projects.
>
> One can argue in LLMs, RLHF preference fine-tuning is also not an authoritative source of truth as humans have biased preferences. Yet, there is still value in tuning with these reward models.
>
> Therefore, we believe the term oracle is reasonable, even if they provide approximations to the "true" value. The most important aspect is that one is aware of the limitations of the oracle, which we explicitly acknowledge in our conclusions section.
>
> ## **4. Why use $B_{ref}$ instead of entire block space? Large block space = sparse reward. Small block subset = reward denser but worse generalization**
>
> We use $B_{enf}$ (we assume the reviewer meant this, please correct us if this is not true) because we are tackling **constrained synthesizability**. We kindly refer to our response to **Reviewer 5JW4**.
>
> Contrary to what the reviewer suggests, **a large block space actually makes the reward more dense, rather than more sparse.** More building blocks generally means more molecules can be “synthesized” from them. More synthesizable molecules means more molecules will have a reward as TANGO assigns a reward to all synthesizable molecules **(Fig. 2)**. Non-synthesizable molecules receive 0 reward.
>
> The reviewer is correct that **enforcing** the presence of specific blocks may restrict generalization (in the sense of what is synthesizable). ***But this is exactly our problem setting and why constrained synthesizability is challenging.***
>
> We recognize that the reviewer may have wrote this comment due to interpreting our framework as RL from scratch instead of post-training. However, we wanted to provide an answer to help clarify our work.
>
> ## **5. TANGO resembles PRM but we should use outcome reward model**
>
> We would like to clarify that TANGO **is** an outcome reward model already. We reference Fig. 2 in our main text. Every synthesizable molecule has a synthesis pathway. TANGO measures the max similarity between every molecule node and the set of enforced blocks ($B_{enf}$). There are no intermediate rewards in our framework. Generating molecules in token space only rewards the final molecule to ensure the molecule is *chemically valid*. Therefore, TANGO returns an outcome reward given the final molecule and its synthesis pathway.
>
> ## **6. Use post-training instead of RL from scratch**
>
> We are already using post-training. Our model is pre-trained on PubChem which is a dataset of bioactive molecules. This pre-training procedure is detailed in Appendix A. This pre-training is performed once. Next, we post-train the model using RL. This is akin to LLM pre-training and RLHF fine-tuning.
>
> We hope to have clarified our work and thank the reviewer for their time. Please let us know if there are additional questions!
>
>
> [1] https://arxiv.org/abs/2206.12411
>
> [2] https://pubs.rsc.org/en/content/articlelanding/2024/md/d3md00651d

---

> > ### Comment · Reviewer_337C · 2025-04-02
> >
> > 1. In fact, the paper [1] already adopts a subset of the full reaction template set. While your work focuses on reducing the size of the starting material set, I believe both approaches share a common goal: constraining the search space to improve the convergence of reinforcement learning loss.
> >
> > [1] Amortized Tree Generation for Bottom-up Synthesis Planning and Synthesizable Molecular Design
> >
> > 2. I believe the term oracle is often abused in the drug discovery community. In reality, single-step retrosynthesis is far from reliable, and referring to it as an "oracle" is, in my opinion, not rigorous.
> >
> > 3. I recommend reviewing the DeepSeek R1 paper, which uses an outcome reward rather than a process reward. This design choice is due to the large sampling space of LLMs and the resulting sparsity of useful rewards. Previous work relied on process rewards to provide more frequent feedback. However, with a strong pretrained model, outcome-based rewards can be used directly. Your paper takes an alternative route by reducing the size of the sampling space to improve reward signal quality, but this comes at the cost of reduced generalization.
> >
> > 4. The outcome reward used in DeepSeek directly reflects the feasibility of a synthesis path. It is a binary signal indicating whether the path is valid or not. I encourage you to take a closer look at the DeepSeek R1 paper for details.
> >
> > 5. Apologies for initially overlooking this point—but since your approach is based on a pretrained model, you should theoretically already have a strong initialization. As such, it may not be necessary to restrict the size of the starting material space purely to improve convergence.
> >
> >
> > ----
> > update
> >
> >
> > The reviewer didn't address my concern. I don’t believe it is reasonable to restrict the block size to such a small value, as different products typically require different building blocks for synthesis.
> >
> > ---
> > Update
> >
> > Additionally, some of the terminology used may mislead the community.

---

> > > ### Author Response · Authors · 2025-04-02
> > >
> > > Thank you to the reviewer for engaging with us! We respectively suggest there are fundamental aspects of our work that are being misinterpreted. We first define our problem setting again and then respond to the reviewer point-by-point.
> > >
> > > ## **What is constrained synthesizability?**
> > >
> > > Our goal is to generate molecules that satisfy multi-parameter optimization **while additionally** satisfying the following 2 properties:
> > >
> > > 1. Synthesizable
> > > 2. **Constrained synthesizable**
> > >
> > > ***2*** is more difficult than ***1*** and is the central problem we are tackling. We first describe ***1*** and then extend to ***2***. Note that the set of reaction rules is **fixed** in both ***1*** and ***2*** .
> > >
> > > "Synthesizable" means there is ***any*** synthesis pathway to a molecule, i.e., **commercially available chemicals (denoted $B_{stock}$)** can be assembled into target molecule. This is what is done in all existing works, including [1] cited by the reviewer.
> > >
> > > We now define $B_{enf} \subset B_{stock}$ and $|B_{enf}| \in [5, 10, 100]$ which represents the set of enforced blocks. Note that $|B_{enf}|$ << $|B_{stock}| (17.7M) $. **Constrained synthesizable** means that the synthesis pathway uses chemicals from **both** $B_{enf}$ and $B_{stock}$. ***A molecule that is synthesizable does not mean that it can be synthesized **while** incorporating $B_{enf}$.***
> > >
> > > We draw an analogy to DeepSeek R1. In their definition of **Accuracy reward** on page 6 of [2], it is stated:
> > >
> > > ***“the model is required to provide the final answer in a specified format (e.g., within a box), enabling reliable rule-based verification of correctness.”***
> > >
> > > We imagine for a second that there is a magic function that can assess correctness *regardless* of output format. Then getting the correct answer (***synthesizable***) is easier than getting the answer correct **and** in the specified format (***constrained synthesizable***).  Getting the correct answer **does not** imply the correct format. Similarly, in our problem setting, a molecule that has a synthesis pathway **does not** mean the leaf nodes incorporate the enforced blocks. We hope this conveys that our problem setting is different and more difficult than general synthesizability.
> > >
> > > ## **Responding to the reviewer’s points**
> > >
> > > **1. and 5.** We are not reducing the size of the starting material set, we continue to use the full **commercially available** set of 17.7M chemicals. We only want that the synthesis pathway incorporates at least 1 enforced block ***amongst*** these 17.7M commercially available chemicals.
> > >
> > > \
> > > **2.** We respect the reviewer’s perspective and can change our terminology to “reward model/function” in the future version.
> > >
> > > \
> > > **3.** DeepSeek R1 = DeepSeek-V3-Base + GRPO to *incentivize* reasoning [2]. They apply
> > > **Accuracy** and **Format** rewards which assess model **output** rather than the generation process, as the reviewer also states.
> > >
> > > **Our framework is the same thing**. We take the pre-trained “PubChem Base Model” (pre-trained on 88M molecules which is a big dataset in chemistry) + RL to *incentivize* the generation of molecules that satisfy constrained synthesizability. In the exact same manner, our model has no inductive biases, it is being guided **only** by the **outcome reward (TANGO)**.
> > >
> > >  ***We are not reducing the size of the sampling space to improve reward signal. The sampling space is not restricted at all. The model can generate any molecules it wants and the outcome reward guides the optimization. This is exactly the same as DeepSeek R1’s workflow.***
> > >
> > > \
> > > **4.** The reviewer states that the binary outcome reward **directly reflects the feasibility of a synthesis path.** However, we show that binary outcome can be sparse and brute-forcing constrained synthesizability with binary rewards is difficult. We refer to **Table 5 first row in the Appendix** showing “Brute-force” (binary outcome reward) is unstable. Our contribution is formulating the TANGO reward that is **still an outcome reward** but makes the reward **dense**. A molecule that is “synthesizable” does not imply“constrained synthesizable”, yet there is meaningful information that can be learned from this. We refer to **Fig. 2 in our main text** which shows **how** a non-zero reward can be returned even if a molecule’s synthesis pathway does not contain an enforced block.
> > >
> > > ---
> > >
> > > **Update:** We address "block restriction". 5, 10, or 100 enforced blocks and **17.7 million** general blocks. Suppose step 1 uses an enforced block -- barring chemical incompatibilities, step 2 can choose from up to 17.7M blocks. The next step can choose again from 17.7M blocks, etc. Therefore, the presence of an enforced block still allows for an **enormous synthesizable space** (**See our response to Reviewer ogSL** where existing works use **much** smaller building block sizes).
> > >
> > >
> > > \
> > > \
> > > [1] https://arxiv.org/abs/2110.06389
> > >
> > > [2] https://arxiv.org/abs/2501.12948

---

### Official Review · Reviewer_ogSL · 2025-03-03

**Overall Recommendation:** 1

**Summary:**

Through this paper, the authors propose TANimoto Group Overlap (TANGO), a reward function for constrained synthesizable molecule generation based on reinforcement learning. The proposed TANGO augments molecular generative models to directly optimize for constrained synthesizability while simultaneously optimizing for other properties relevant to drug discovery.

### **Update after rebuttal**
Thank you for the authors for the rebuttal. However, my concerns are not completely resolved. An approach like Synformer, which performs the nearest neighbor search with the molecule fingerprints within a given set of building blocks, has some generalizability even if the BB set is different in the training and inference phases. And since Synformer does not need to predict intermediate molecules, but only the earliest BBs, there is no need to choose from which step to adopt original BBs and from which step to adopt enforced BBs. All BB can be selected from the enforced set. Overall, I think the problem the paper is solving is already possible as a trivial extension of current methods, or at least, a comparison with a trivial extension of existing methods is essential, but not available.

**Claims And Evidence:**

The claims made in the submission are supported by evidence.

**Essential References Not Discussed:**

The submission covers essential references.

**Experimental Designs Or Analyses:**

There is no comparison with existing methods in this paper. In existing synthesizable molecular design literatures [1-4], a very popular strategy is to generate Morgan fingerprints of building blocks and then perform the nearest neighbor search on a library of predefined building blocks. I think that one can easily perform constrained synthesizable molecular design by restricting the library for performing nearest neighbor search to the enforced building blocks. What is the advantage of this work compared to this approach? And I strongly think that extensive further comparison with existing works is needed.

---
**References:**

[1] Gao et al., Amortized tree generation for bottom-up synthesis planning and synthesizable molecular design, ICLR, 2022.

[2] Gao et al., Generative artificial intelligence for navigating synthesizable chemical space, arXiv, 2024.

[3] Cretu et al., SynFlowNet: design of diverse and novel molecules with synthesis constraints, ICLR, 2025.

[4] Sun et al., Procedural synthesis of synthesizable molecules, ICLR, 2025.

**Methods And Evaluation Criteria:**

The proposed methods and evaluation criteria make sense for the proposed problem.

**Other Comments Or Suggestions:**

--

**Other Strengths And Weaknesses:**

This work addressed a new problem in the ML domain: constrained synthesizability, but at the same time, it was not backed up with sufficient explanations and examples of why this is an important real-world problem.

**Questions For Authors:**

--

**Relation To Broader Scientific Literature:**

My biggest concern for this work is the limited novelty, both conceptually and technically. The proposed framework is be a straighforward integration of a retrosynthesis prediction model (i.e., Syntheseus [5]) and a molecular generative model (i.e., Saturn [6]). Overall, it is more like a heuristic technique rather than an ML algorithm. I realize that combining two already existing methodologies does not always result in low contribution work, but in this case it's a straightforward solution and I do not think there are any particular challenges in combining the two that this work solved.

---
**References:**

[5] Maziarz et al., Re-evaluating retrosynthesis algorithms with syntheseus, NeurIPS AI4Science Workshop, 2023.

[6] Guo et al., Saturn: sample-efficient generative molecular design using memory manipulation, arXiv, 2024.

**Theoretical Claims:**

There are no theoretical claims in this paper.

---

> ### Author Rebuttal · Authors · 2025-03-29
>
> Thank you to the reviewer for their feedback and for the opportunity to clarify our work.
>
> `FP` = fingerprint
>
> `NN` = nearest-neighbor
>
> ## **1. Why is constrained synthesizability important?**
>
> We kindly refer to our response to **Reviewer 5JW4**.
>
> ## **2. Why existing synthesizable design works cannot be easily adapted to this problem setting**
>
> We first highlight characteristics of recent synthesizable design works and then discuss why adapting these models for constrained synthesizability is non-trivial.
>
> **SynNet** [1]
> * 147,505 blocks
> * `FP` + `NN` to select blocks
> * Pre-training dataset prepared by sampling blocks-reactions to generate synthetic trees
>
> **SynFormer** [2]:
> * Diffusion to select blocks, allowing scaling to longer `FP`
> * 223,224 blocks
> * Pre-training dataset prepared by sampling blocks-reactions to generate postfix notations [3] of synthesis
>
> **RGFN** [4]:
> * Either 350 or 8350 blocks
> * Memory constraints limiting scaling to larger block sets (see Appendix B.5 and Appendix D.3 in [6])
>
> **SynFlowNet** [5]:
> * Scaling to **up to** (as stated in the paper) 200k blocks but most experiments are run with 10,000 blocks
> * Uses masking to enforce compatible blocks and reactions
> * Memory constraints limiting scaling to larger block sets (see Appendix B.5 and Appendix D.3 in [6])
>
> **RxnFlow** [6]:
> * Scale up to 1.2M blocks
> * Non-hierarchical MDP formulation compared to hierarchical in RGFN [4] and SynFlowNet [5] allowing adaptation to changing blocks
>
> In all methods above, blocks are selected either by `FP` `NN` or softmax sampled. Constrained synthesizability means that a small set (in our work we considered sizes of 5, 10, 100) of enforced blocks are **present** in the synthesis pathways. With `FP`, the reviewer suggests doing `NN` search on the enforced blocks. However, it is non-trivial to decide at which step of the synthesis pathway to do this in the general case, e.g., choose a "normal" block at step $t$ and then force step $t+1$ to choose an enforced block? What if the enforced blocks are incompatible with the reaction? One may need to encode explicit biases into the generation process which TANGO overcomes by *incentivizing* the learning process. One could *mask* the actions to guarantee enforced blocks, but the same consideration remains in *when* to do this and what to do for block-reaction incompatibility. For SynNet [1] and SynFormer [2] which require pre-training with enumerated “pseudo” routes, should one generate more routes with the enforced blocks to increase their sampling probability? Would this diminish the model’s ability to generate diverse molecules? We believe these are open questions for model development.
>
> We next highlight a scalability limitation. We used 17.7M blocks which is 2-3 orders of magnitude larger than existing works which need to change model dimensions and/or more memory to increase block set. We can scale to 100M blocks since we only need to store the SMILES. **But why use so many blocks?** It is not *unreasonable* to consider *any and all* blocks that are commercially available. Our framework also allows freely changing block sets while the existing methods need to re-initialize (see [7] which applies Saturn using 5 distinct block stocks **without** re-training).
>
> ***TANGO guides Saturn, which is a model with no synthesizability inductive biases, to optimize for constrained synthesizability. TANGO is general and can augment these existing models to do this and designing this function is the contribution of our work.*** However, we note that it may still not be straightforward as the lower sample efficiency of GFlowNet models can make this computationally prohibitive, i.e., constrained synthesizability alone is not enough, the molecules' properties must also be optimized (see [8] for a benchmark where GFlowNets are ranked 16/25 for sample efficiency and [7] for a Saturn/GFlowNet comparison). See also Appendix A.5 in SynFlowNet [5] showing that its sample efficiency is lower than REINVENT [9], which Saturn significantly outperforms (see Table 3 in [10]). **Lastly, we report docking scores across thresholds to show that our framework performs multi-parameter optimization.**
>
> ***We now have results showing TANGO can augment GraphGA (genetic algorithm) for constrained synthesizability and will update the manuscript when we are able to do so.***
>
> We are eager to continue discussion and hope we clarified some points. Please let us know if there are follow-up questions!
>
> [1] https://openreview.net/forum?id=FRxhHdnxt1
>
> [2] https://arxiv.org/abs/2410.03494
>
> [3] https://openreview.net/forum?id=scFlbJQdm1
>
> [4] https://openreview.net/forum?id=hpvJwmzEHX
>
> [5] https://openreview.net/forum?id=uvHmnahyp1
>
> [6] https://openreview.net/forum?id=pB1XSj2y4X
>
> [7] https://pubs.rsc.org/en/content/articlehtml/2025/sc/d5sc01476j
>
> [8] https://arxiv.org/abs/2206.12411
>
> [9] https://jcheminf.biomedcentral.com/articles/10.1186/s13321-017-0235-x
>
> [10] https://arxiv.org/abs/2405.17066

---

### Official Review · Reviewer_yDvi · 2025-03-09

**Overall Recommendation:** 4

**Summary:**

The paper "It Takes Two to Tango" introduces a new approach to generative molecular design that explicitly optimizes for synthesizability under real-world constraints. The key problem is existing molecular generative models often optimize for molecular properties (such as drug-likeness or docking scores) but fail to ensure that the generated molecules can be synthesized. This is especially relevant when specific starting materials or intermediates must be used.

To tackle this, the authors propose TANGO (TANimoto Group Overlap), a reward function designed to transform the sparse binary signal from retrosynthesis models into a more dense signal. By incorporating Tanimoto similarity, functional group overlap, and fuzzy matching, TANGO helps reinforcement learning models navigate chemical space more effectively. The generative model used is Saturn, a reinforcement learning-based autoregressive model operating on SMILES sequences. It is coupled with the MEGAN retrosynthesis model and the Retro search algorithm* to ensure that generated molecules are both synthesizable and optimized for specific chemical properties.

**Claims And Evidence:**

Overall, most claims in the paper are well-supported by experimental results, particularly the effectiveness of the TANGO reward function in improving reinforcement learning for constrained molecular generation. The authors provide strong evidence that their model can enforce synthesizability constraints while optimizing molecular properties, with extensive ablation studies confirming the advantage of TANGO over simpler similarity-based rewards. The results demonstrate that the method works across different synthesis constraints and successfully generates molecules that balance synthesizability with desired drug-like properties.

However, the discussion on true synthesizability is limited—since retrosynthesis models have imperfections, it would help to assess whether the generated molecules are chemically feasible beyond just passing retrosynthesis predictions. Adding a more in-depth discussion of these limitations would make the paper stronger.

**Essential References Not Discussed:**

No

**Experimental Designs Or Analyses:**

No issues

**Methods And Evaluation Criteria:**

The proposed methods and evaluation criteria are mostly well-aligned with the problem of constrained synthesizability in generative molecular design.

**Other Comments Or Suggestions:**

Great paper!

**Other Strengths And Weaknesses:**

Addressed before

**Questions For Authors:**

No

**Relation To Broader Scientific Literature:**

The paper situates itself within the broader scientific literature on generative molecular design, retrosynthesis modeling, and reinforcement learning for molecule generation. It builds upon prior work in synthesizability-constrained molecular design and reinforcement learning-based molecular optimization while introducing a novel approach to enforcing chemical constraints directly within the generative model.

**Theoretical Claims:**

No issues

---

> ### Author Rebuttal · Authors · 2025-03-29
>
> Thank you to the reviewer for their positive assessment of our work. The point about true synthesizability is very important to us, as experimental validation is always the end goal of generative design.
>
> As we are unable to update the manuscript version at this time, we wanted to provide more discussion in our response here. The major limitation of retrosynthesis models is that proposed pathways might suffer from general feasibility (does the reaction *actually* proceed?) and selectivity problems - regioselectivity, i.e., does the reaction proceed at exactly the position we want? And stereoselectivity, i.e., does the reaction yield the correct enantiomer as the major product? To this end, reaction feasibility/selectivity in retrosynthesis has been explored by integrating quantum chemistry information either through simulations [1] or the use of machine learning force fields [2] (we cite a recent example each and a recent review here [3]). Effectively and efficiently leveraging quantum chemistry information for reaction feasibility is an open research problem. However, going forward, more capable retrosynthesis models would immediately benefit our framework.
>
> Finally, in the reaction pathways shown in the main text, amide coupling reactions are predominantly present (partially owing to how common they are). In general, this is a relatively robust reaction. In the updated manuscript, we will explicitly comment on the feasibility of the reactions.
>
> [1] https://chemrxiv.org/engage/chemrxiv/article-details/671f791c1fb27ce124d5c98c
>
> [2] https://chemrxiv.org/engage/chemrxiv/article-details/67d7b7f7fa469535b97c021a
>
> [3] https://pubs.rsc.org/en/content/articlelanding/2025/sc/d5sc00541h

---

### Official Review · Reviewer_5JW4 · 2025-03-14

**Overall Recommendation:** 3

**Summary:**

This paper focuses on the challenge of directly optimizing for constrained synthesizability in generative molecular design. Controlling the synthesizability of generated molecules is crucial for closed-loop discovery and robotic synthesis automation. Existing methods have limitations, and there is a lack of molecular generative models that can enforce specific building blocks in synthesis routes. The authors propose a novel reward function named TANimoto Group Overlap (TANGO) to address this issue. The experiment shows that the TANGO reward function can guide a general-purpose molecular generative model to optimize for constrained synthesizability and perform MPO simultaneously.

**Claims And Evidence:**

1. For Line 37 "Our framework is the first generative approach to successfully address constrained synthesizability". There might be an overclaim in this field. For example, some crystal material generation methods already investigated constrained synthesizability from the perspective of structure stability.

2. Line 53:  However, to date, there are no molecular generative models that can enforce specific building blocks in the proposed routes.
I did not find a detailed comparison between the existing works and the proposed method in this direction.

"More recently, constrained retrosynthesis algorithms have been proposed", could you provide more explanation for this?

**Essential References Not Discussed:**

Not found.

**Experimental Designs Or Analyses:**

Yes, I did not find impactful issues for the experimental designs.

**Methods And Evaluation Criteria:**

The experimental setup, including the choice of the drug discovery case study (optimizing docking scores against a specific protease and QED values) make sense to the overall goal of generating useful molecules. The metrics used, such as Non-solved, Solved (Enforced), docking scores, QED values, and the number of reaction steps, are appropriate for evaluating the performance of the model in terms of synthesizability and multi-parameter optimization.

**Other Comments Or Suggestions:**

1. The font in Figure 1 is too small.

**Other Strengths And Weaknesses:**

1. The core idea of TANGO  and the motivation is not clear.  In Figure 1,  it's not clear to find the reason that simultaneously uses enforced building blocks and optimizing other properties.

**Questions For Authors:**

Except the drug discovery, what's the potential impact on the chemistry and meterial science?

**Relation To Broader Scientific Literature:**

This work proposed a new framework for onstrained synthesizability in generative molecular design, and has potential impact on the chemistry and meterial science, as well as the drug discovery.

**Theoretical Claims:**

There are no Theoretical Claims.

---

> ### Author Rebuttal · Authors · 2025-03-29
>
> We thank the reviewer for their feedback and an opportunity to clarify our problem setting.
>
> ## **Generative approach for constrained synthesizability. Why is this useful?**
>
> We answer all questions from the reviewer in this single response since they are related.
>
> **What is the problem of constrained synthesizability?**
>
> *Synthesizing* molecules can be thought of as stitching together “building blocks” (commercially available chemicals) with reactions. We will refer to this set of building blocks as $B_{stock}$. Constrained synthesizability extends this problem setting to also enforce that a **specific** set of building blocks is used in the synthesis and denote this $B_{enforced}$. We emphasize that there are **much fewer** $B_{enforced}$: $|B_{stock}|$ is 17.7 million while we used $|B_{enforced}|$ $\in$ [5, 10, 100]. This is a significantly harder problem because the model must generate molecules that can be synthesized by using **both** $|B_{enforced}|$ and $|B_{stock}|$ and blocks cannot be freely combined together as they must be chemically compatible. ***TANGO enables a completely unconstrained generative model to learn how to satisfy this constraint using reinforcement learning, while performing multi-parameter optimization of other properties***.
>
> **Why should we care about enforcing specific building blocks?**
>
> Building blocks have variable costs. One can envision enforcing the use of the cheapest set of blocks from a commercial vendor to manage costs.
>
> We may want to re-purpose certain blocks into useful molecules. For example in [1], waste molecules from industrial processes are repurposed into medicines. By definition, this is constrained synthesizability because one wants to use these waste molecules ($B_{enforced}$) **together** with general chemicals that can be purchased ($B_{stock}$). One could conceive using TANGO to repurpose waste into *de novo* molecules.
>
> In drug discovery, we often want to generate molecules that share a common scaffold and diversify from this scaffold. In Fig. 3 right panel, we show this capability explicitly where a 1-step reaction from an enforced scaffold forms a new molecule with improved properties. An important capability of TANGO here is that > 1 block can be considered simultaneously and it is not uncommon that one is interested in multiple scaffolds. There are existing works that can enforce a specific scaffold with pre-defined exit vectors to attach new chemical groups to but TANGO is much more general.
>
> **Where is constrained synthesizability useful beyond drug discovery?**
>
> In this paper, we focused on organic molecules for drug discovery. Organic molecules encompass many **classes** of matter, for example organocatalyst design [3] and functional materials design like semiconductors [4]. TANGO could be extended to tackle these problems.
>
> **Overclaim of first generative approach and no discussion of existing works**
>
> In section “Synthesizability-constrained Molecular Generation”, we cited many works that have indeed tackled synthesizability in generative models. However, our problem setting is **constrained synthesizability** for which the existing works do not tackle. More specifically, we generate molecules with an **explicit synthesis pathway** (step-by-step recipe to make the molecule) that also incorporates specific blocks. In Appendix F, we show examples of these synthesis pathways and put a box around the specific enforced block used. While we operate in the organic molecules space, we are also, to the best of our knowledge, not aware of crystal (inorganic) generation works with constrained synthesizability.
>
> For detailed discussions of why existing works cannot easily be extended to constrained synthesizability, we kindly refer the reviewer to our response to **Reviewer ogSL**.
>
> **More recently, constrained retrosynthesis algorithms have been proposed. More details about this**
>
> [5] is a specific recently proposed constrained retrosynthesis algorithm. **Retrosynthesis is a different problem setting to ours**: retrosynthesis involves generating a synthesis pathway **given a target molecule**. We do not have a “target molecule” in the same sense, as TANGO is guiding the model to **generate** molecules that have a synthesis pathway containing the enforced blocks **and** containing optimized properties. Furthermore, in [5], the algorithm is trying to propose a synthesis pathway that incorporates a **single** pre-defined block. TANGO can generalize to a **set** of blocks, shown by the results when we enforced 5, 10, 100 blocks.
>
> Thank you to the reviewer again and we would be eager to continue discussion!
>
>
> [1] https://www.nature.com/articles/s41586-022-04503-9
>
> [2] https://pubs.acs.org/doi/10.1021/acs.jcim.1c00469
>
> [3] https://onlinelibrary.wiley.com/doi/full/10.1002/anie.202218565
>
> [4] https://pubs.rsc.org/en/content/articlelanding/2020/nr/c9nr10687a
>
> [5] https://openreview.net/forum?id=LJNqVIKSCr

---

### Decision · Program_Chairs · 2025-05-01

**Decision:**

Reject

**Comment:**

This paper proposes a framework for constrained synthesizability in molecular generative models. Specifrically, a reward function is used to provide dense RL feedback to guide a generative model in a multi objective optimization setting to optimize molecules while keeping them syntheizable.

On the positive side, the reviewers have acknowledged the importance of the problem, revelavance of the proposed method, and the design of dense RL feedback. However, they have raised serious concerns around the limitation of retrosynthesis oracles, lack of head-to-head comparisons with straightforward baselines or previous state of the art methods, and different reward designs choices. Given these concerns the paper is not ready for publication at ICML.